# FedGame: A Game-Theoretic Defense against Backdoor Attacks in Federated Learning

**Jinyuan Jia** *
The Pennsylvania State University
jinyuan@psu.edu

**Zhuowen Yuan** *
UIUC
zhuowen3@illinois.edu

**Dinuka Sahabandu**
University of Washington
sdinuka@uw.edu

**Luyao Niu**
University of Washington
luyaoniu@uw.edu

**Arezoo Rajabi**
University of Washington
rajabia@uw.edu

**Bhaskar Ramasubramanian**
Western Washington University
ramasub@wwu.edu

**Bo Li**
UIUC
lbo@illinois.edu

**Radha Poovendran**
University of Washington
rp3@uw.edu

## Abstract

Federated learning (FL) provides a distributed training paradigm where multiple clients can jointly train a global model without sharing their local data. However, recent studies have shown that FL offers an additional surface for backdoor attacks. For instance, an attacker can compromise a subset of clients and thus corrupt the global model to misclassify an input with a backdoor trigger as the adversarial target. Existing defenses for FL against backdoor attacks usually detect and exclude the corrupted information from the compromised clients based on a *static* attacker model. However, such defenses are inadequate against *dynamic* attackers who strategically adapt their attack strategies. To bridge this gap, we model the strategic interactions between the defender and dynamic attackers as a minimax game. Based on the analysis of the game, we design an interactive defense mechanism FedGame. We prove that under mild assumptions, the global model trained with FedGame under backdoor attacks is close to that trained without attacks. Empirically, we compare FedGame with multiple *state-of-the-art* baselines on several benchmark datasets under various attacks. We show that FedGame can effectively defend against strategic attackers and achieves significantly higher robustness than baselines. Our code is available at: https://github.com/AI-secure/FedGame.

## 1 Introduction

Federated learning (FL) [21, 28] aims to train a global model over training data that are distributed across multiple clients (e.g., mobile phones, IoT devices) in an iterative manner. In each communication round, a cloud server shares its global model with selected clients. Then, each selected client uses the global model to initialize its local model, utilizes its local training dataset to train the local model, and sends the local model update to the server. Finally, the server uses an aggregation rule to integrate local model updates from clients to update its global model.

Due to the distributed nature of FL, many studies [3, 4, 5, 21, 44] have shown that FL is vulnerable to backdoor attacks [11, 18, 26, 27, 30, 33, 37, 40, 41, 55, 59]. For instance, an attacker can compromise

---

*Equal contribution.

a subset of clients and manipulate their local training datasets to corrupt the global model such that it predicts an attacker-chosen target class for any inputs embedded with a backdoor trigger [3]. To defend against such backdoor attacks, many defenses [7, 32, 34, 36, 39, 51] have been proposed (see Section 2 for details). However, all those defenses consider a *static* attack model where an attacker sticks to a fixed strategy and does not adapt its attack strategies. As a result, they are less effective under adaptive attacks, e.g., Wang et al. [44] showed that defenses in [6, 39] can be bypassed by appropriately designed attacks. While the vulnerability of the FL against dynamic or adaptive attack is known, dynamic defense has not been well-studied yet.

In this work, we propose FedGame, a game-theoretic defense against backdoor attacks in FL. Specifically, we formulate FedGame as a minimax game between the server (defender) and attacker, enabling them to adapt their defense and attack strategies strategically. While the server has no prior knowledge about which client is compromised, our key idea is that the server can compute a *genuine score* for each client, which should be large (or small) if the client is benign (or compromised) in each communication round. The genuine score serves as a weight for the local model update of the client when used to update the global model. The goal of the defender is to minimize the genuine scores for compromised clients and maximize them for benign ones. To solve the resulting minimax game for the defender, we follow a three-step process consisting of 1) building an auxiliary global model, 2) exploiting the auxiliary global model to reverse engineer a backdoor trigger and target class, and 3) testing whether the local model of a client will predict an input embedded with the reverse engineered backdoor trigger as the target class to compute a genuine score for the client. Based on the deployed defense, the goal of the attacker is to optimize its attack strategy by maximizing the effectiveness of the backdoor attack while remaining stealthy. Our key observation is that the attack effectiveness is determined by two factors: the genuine score and the local model of the client. We optimize the attack strategy with respect to those two factors to maximize the effectiveness of backdoor attacks against dynamic defense.

We perform both theoretical analysis and empirical evaluations for FedGame. Theoretically, we prove that the global model trained with our defense under backdoor attacks is close to that trained without attacks (measured by Euclidean distance of model parameters). Empirically, we perform comprehensive evaluations. In particular, we evaluate FedGame on benchmark datasets to demonstrate its effectiveness under *state-of-the-art* backdoor attacks [3, 54, 59]. Moreover, we compare it with six *state-of-the-art* baselines [6, 7, 36, 39, 56]. Our results indicate that FedGame outperforms them by a significant margin (e.g., the attack success rate for FedGame is less than 13% while it is greater than 95% for those baselines on CIFAR10 under Scaling attack [3]). We also perform comprehensive ablation studies and evaluate FedGame against adaptive attacks. Our results show FedGame is consistently effective. Our key contributions are as follows:

- We propose FedGame, the first game-theoretic defense against dynamic backdoor attacks to FL.

- We provide theoretical guarantees of FedGame. We show that the global model trained with FedGame under backdoor attacks is close to that without attacks.

- We perform a systematic evaluation of FedGame on benchmark datasets and demonstrate that FedGame significantly outperforms state-of-the-art baselines.

## 2   Related Work

**Backdoor Attacks in FL.**   In backdoor attacks against FL [3, 4, 5, 31, 44, 50, 59], an attacker aims to make a global model predict a target class for any input embedded with a backdoor trigger via compromised clients. For instance, Bagdasaryan et al. [3] proposed the scaling attack in which an attacker first uses a mix of backdoored and clean training examples to train its local model and then scales the local model update by a factor before sending it to the server. In our work, we will leverage *state-of-the-art* attacks [3, 54, 59] to perform strategic backdoor attacks against our defense.

**Defenses against Backdoor Attacks in FL.**   Many defenses [39, 7, 34, 51, 36, 32, 57, 9] were proposed to mitigate backdoor attacks in FL. For instance, Sun et al. [39] proposed norm-clipping, which clips the norm of the local model update of a client to a certain threshold. Some work extended differential privacy [15, 1, 29] to mitigate backdoor attacks to federated learning. The idea is to clip the local model update and add Gaussian noise. Cao et al. [7] proposed FLTrust, which leveraged the similarity of the local model update of a client with that computed by the server itself on its

clean dataset. DeepSight [36] performs clustering on local models and then filters out those from compromised clients. By design, DeepSight is ineffective when the fraction of compromised clients is larger than 50%. Other defenses include Byzantine-robust FL methods such as Krum [6], Trimmed Mean [56], and Median [56]. However, all of those defenses only consider a static attacker model. As a result, they become less effective against dynamic attackers who strategically adapt their attack strategies.

Another line of research focuses on detecting compromised clients [24, 58]. As those defenses need to collect many local model updates from clients to make confident detection, the global model may already be backdoored before those clients are detected. Some recent studies [8, 53, 57, 10] proposed certified defenses against compromised clients. However, they can only tolerate a moderate fraction of compromised clients (e.g., less than 10%) as shown in their results. In contrast, FedGame can tolerate a much higher fraction (e.g., 80%) of compromised clients.

In this work, we aim to prevent backdoor attacks to federated learning, i.e., our goal is to train a robust global model under compromised clients. The server could also utilize post-training defenses [43, 51] to remove the backdoor in a backdoored global model. Prevention-based defense could be combined with those post-training defenses to form a defense-in-depth. We note that the defense proposed in [43] requires the server to own some clean samples that have the same distribution as the local training data of clients. The defense proposed in [51] prunes filters based on pruning sequences collected from clients and thus is less effective when the fraction of compromised clients is large.

## 3 Background and Threat Model

**Federated Learning.** Let $\mathcal{S}$ denote the set of clients and $\mathcal{D}_i$ denote the local training dataset of client $i \in \mathcal{S}$. In the $t^{th}$ communication round, the server first sends the current global model $\Theta^t$ to the selected clients. Then, each selected client $i$ trains a local model (denoted as $\Theta_i^t$) by fine-tuning the global model $\Theta^t$ using its local training dataset $\mathcal{D}_i$. For simplicity, we use $\mathbf{z} = (\mathbf{x}, y)$ to denote a training example in $\mathcal{D}_i$, where $\mathbf{x}$ is the training input (e.g., an image) and $y$ is its ground truth label. Given $\mathcal{D}_i$ and the global model $\Theta^t$, the local objective is defined as $\mathcal{L}(\mathcal{D}_i; \Theta^t) = \frac{1}{|\mathcal{D}_i|} \sum_{\mathbf{z} \in \mathcal{D}_i} \ell(\mathbf{z}; \Theta^t)$ where $\ell$ is a loss function (e.g., cross-entropy loss). The client can use gradient descent to update its local model based on the global model and its local training dataset, i.e., $\Theta_i^t = \Theta^t - \eta_l \frac{\partial \mathcal{L}(\mathcal{D}_i; \Theta^t)}{\partial \Theta^t}$, where $\eta_l$ is the local learning rate. Note that stochastic gradient descent can also be used in practice. After that, the client sends $g_i^t = \Theta_i^t - \Theta^t$ (called *local model update*) to the server. Note that it is equivalent for the client to send a local model or local model update to the server as $\Theta_i^t = \Theta^t + g_i^t$. After receiving the local model updates from all clients, the server can aggregate them based on an aggregation rule $\mathcal{R}$ (e.g., FedAvg [28]) to update its global model, i.e., we have $\Theta^{t+1} = \Theta^t + \eta \mathcal{R}(g_1^t, g_2^t, \cdots, g_{|\mathcal{S}|}^t)$, where $|\mathcal{S}|$ represents the number of clients and $\eta$ is the learning rate of the global model.

**Threat Model.** We consider attacks proposed in previous work [3, 54, 59]. We assume an attacker can compromise a set of clients (denoted as $\mathcal{S}_a$). To perform backdoor attacks, the attacker first selects an arbitrary backdoor trigger $\delta$ and a target class $y^{tc}$. For each client $i \in \mathcal{S}_a$ in the $t^{th}$ ($t = 1, 2, \dots$) communication round, the attacker can choose an arbitrary fraction (denoted as $r_i^t$) of training examples from the local training dataset of the client, embed the backdoor trigger $\delta$ to training inputs, and relabel them as the target class $y^{tc}$. In *state-of-the-art* backdoor attacks [3, 54, 59], the attacker leverages those trigger-embedded training examples to inject the backdoor into the local model of a compromised client to attack the global model.

Following [7], we consider that the server itself has a small clean training dataset (denoted as $\mathcal{D}_s$), which could be collected from the same or different domains of the local training datasets of clients. Note that the server does not have 1) information on compromised clients, and 2) the poisoning ratio $r_i^t$ ($i \in \mathcal{S}_a$), backdoor trigger $\delta$, and target class $y^{tc}$ selected by the attacker. In our threat model, we do not make any other assumptions for the server compared with standard federated learning [28].

## 4 Methodology

**Overview.** We formulate FedGame as a minimax game between the defender and attacker, which enables them to optimize their strategies respectively. In particular, the defender computes a genuine

score for each client in each communication round. The goal of the defender is to maximize the genuine score for a benign client and minimize it for a compromised one. Given the genuine score for each client, we use a weighted average over all the local model updates to update the global model, i.e., we have

$$\Theta^{t+1} = \Theta^t + \eta \frac{1}{\sum_{i \in \mathcal{S}} p_i^t} \sum_{i \in \mathcal{S}} p_i^t g_i^t, \tag{1}$$

where $p_i^t$ is the genuine score for client $i$ in the $t^{th}$ communication round and $\eta$ is the learning rate of the global model. We compute a weighted average of local models because we aim to achieve robustness against attacks while making minimal changes to FedAvg [28], which has been shown to achieve high utility in diverse settings. In our experiments, we empirically show that the genuine scores are almost constantly 0 for malicious clients after a few communication rounds, which indicates that the malicious model updates will not be aggregated at all, reducing Equation 1 to a robust FedAvg against attacks. (See Figure 2(c)(d) in the Appendix for more details.) Based on Equation 1, the effectiveness of the attack is determined by two components: genuine scores and local models of compromised clients. In our framework, the attacker optimizes the tradeoff between these two components to maximize the effectiveness of backdoor attacks.

### 4.1 Game Formulation

**Computing Genuine Scores.** The key challenge for the server to compute genuine scores for clients is that the server can only access their local model updates, i.e., the server can only leverage local models of clients to compute genuine scores. To tackle this issue, we observe that the local model of a compromised client is more likely to predict a trigger-embedded input as the target class. Therefore, the server can first reverse engineer the backdoor trigger $\delta_{re}$ and target class $y_{re}^{tc}$ (which we will discuss more in the next subsection) and then use them to compute the genuine score for each client. Since the client $i$ sends its local model update $g_i^t$ to the server, the server can compute the local model of the client $i$ as $\Theta_i^t = \Theta^t + g_i^t$. With $\Theta_i^t$, the server can compute $p_i^t$ as follows:

$$p_i^t = 1 - \frac{1}{|\mathcal{D}_s|} \sum_{\mathbf{x} \in \mathcal{D}_s} \mathbb{I}(G(\mathbf{x} \oplus \delta_{re}; \Theta_i^t) = y_{re}^{tc}), \tag{2}$$

where $\mathbb{I}$ is an indicator function, $\mathcal{D}_s$ is the clean training dataset of the server, $\mathbf{x} \oplus \delta_{re}$ is a trigger-embedded input, and $G(\mathbf{x} \oplus \delta_{re}; \Theta_i^t)$ represents the label of $\mathbf{x} \oplus \delta_{re}$ predicted by $\Theta_i^t$. Intuitively, the genuine score for client $i$ is small if a large fraction of inputs embedded with the reverse-engineered backdoor trigger is predicted as the target class by its local model. We note that the defender only needs to query local models to compute their genuine scores. Moreover, our defense is still applicable when differential privacy [14, 49] is used to protect the local models.

**Optimization Problem of the Defender.** The server aims to reverse engineer the backdoor trigger $\delta_{re}$ and target class $y_{re}^{tc}$ such that the genuine scores for compromised clients are minimized while those for benign clients are maximized. Formally, we have the following optimization problem:

$$\min_{\delta_{re}, y_{re}^{tc}} \left( \sum_{i \in \mathcal{S}_a} p_i^t - \sum_{j \in \mathcal{S} \setminus \mathcal{S}_a} p_j^t \right). \tag{3}$$

Note that $\mathcal{S}_a$ is the set of malicious clients that is unknown to the server. In the next subsection, we will discuss how to address this challenge to solve the optimization problem.

**Optimization Problem of the Attacker.** The goal of an attacker is to maximize its attack effectiveness. Based on Equation 1, the attacker needs to: 1) maximize the genuine scores for compromised clients while minimizing them for benign ones, i.e., $\max(\sum_{i \in \mathcal{S}_a} p_i^t - \sum_{j \in \mathcal{S} \setminus \mathcal{S}_a} p_j^t)$, and 2) make the local models of compromised clients predict an input embedded with the attacker-chosen backdoor trigger $\delta$ as the target class $y^{tc}$. To perform the backdoor attack in the $t^{\text{th}}$ communication round, the attacker embeds the backdoor to a certain fraction (denoted as $r_i^t$) of training examples in the local training dataset of the client and uses them as data augmentation. Intuitively, a larger $r_i^t$ encourages a higher attack success rate but can potentially result in a lower genuine score. In other words, $r_i^t$ measures a tradeoff between these two factors. Formally, the attacker can solve the following

optimization problem to find the desired tradeoff:

$$\max_{R^t, \delta}(\sum_{i \in \mathcal{S}_a} p_i^t - \sum_{j \in \mathcal{S} \setminus \mathcal{S}_a} p_j^t + \lambda \sum_{i \in \mathcal{S}_a} r_i^t), \tag{4}$$

where $R^t = \{r_i^t \mid i \in \mathcal{S}_a\}$, $\delta$ is the trigger, and $\lambda$ is a hyperparameter to balance the two terms.

**Minimax Game.** Given the optimization problems of the defender and attacker, we have the following minimax game:

$$\min_{\delta_{re}, y_{re}^{tc}} \max_{R^t, \delta}(\sum_{i \in \mathcal{S}_a} p_i^t - \sum_{j \in \mathcal{S} \setminus \mathcal{S}_a} p_j^t + \lambda \sum_{i \in \mathcal{S}_a} r_i^t). \tag{5}$$

Note that $r_i^t$ ($i \in \mathcal{S}_a$) and $\delta$ are chosen by the attacker. Thus, we can add them to the objective function in Equation 3 without influencing its solution given the local model updates of clients. In Section A of Appendix, we provide an intuitive interpretation regarding how to connect our above objective to the ultimate goal of the defender and attacker (the defender aims to obtain a clean model, while the attacker wants the global model to be backdoored).

## 4.2 Solving the Minimax Game by the Defender

The key challenge for the server to solve Equation 5 is that it does not know $\mathcal{S}_a$ (set of compromised clients). To address the challenge, our idea is to construct an *auxiliary global model* based on local models of all clients. Suppose $g_i^t$ is the local model update sent by each client $i \in \mathcal{S}$ to the server. Our auxiliary global model is constructed as follows: $\Theta_a^t = \Theta^t + \frac{1}{|\mathcal{S}|} \sum_{i \in \mathcal{S}} g_i^t$, where $\Theta^t$ is the global model. Our intuition is that such a naively aggregated auxiliary global model is inclined to predict a trigger-embedded input as the target class under backdoor attacks. As a result, given the auxiliary global model, we can use an arbitrary existing method [42, 46, 38, 48] to reverse engineer the backdoor trigger and target class based on it, which enables us to compute genuine scores for clients based on Equation 2. With those genuine scores, we can use Equation 1 to update the global model to protect it from backdoor attacks in every communication round. The complete algorithm of our FedGame is shown in Algorithm 1 of Appendix.

Our framework is compatible with any trigger reverse engineering methods, allowing off-the-shelf incorporation of techniques developed for centralized supervised learning into federated learning. Note that developing a new reverse engineering method is not the focus of this work. Instead, our goal is to formulate the attack and defense against backdoor attacks on federated learning as a minimax game, which enables us to defend against dynamic attacks.

## 4.3 Solving the Minimax Game by the Attacker

The goal of the attacker is to find $r_i^t$ for each client $i \in \mathcal{S}_a$ such that the loss function in Equation 5 is maximized. As the attacker does not know the genuine scores of benign clients, the attacker can find $r_i^t$ to maximize $p_i^t + \lambda r_i^t$ for client $i \in \mathcal{S}_a$ to approximately solve the optimization problem in Equation 5. However, the key challenge is that the attacker does not know the reverse engineered backdoor trigger $\delta_{re}$ and the target class $y_{re}^{tc}$ of the defender to compute the genuine score for the client $i$. In response, the attacker can use the backdoor trigger $\delta$ and target class $y^{tc}$ chosen by itself. Moreover, the attacker reserves a certain fraction (e.g., 10%) of training data from its local training dataset $\mathcal{D}_i$ as the validation dataset (denoted as $\mathcal{D}_i^{rev}$) to find the best $r_i^t$.

**Estimating the Genuine Score Given $r_i^t$.** For a given $r_i^t$, the client $i$ can embed the backdoor to $r_i^t$ fraction of training examples in $\mathcal{D}_i \setminus \mathcal{D}_i^{rev}$ and then use those backdoored training examples to augment $\mathcal{D}_i \setminus \mathcal{D}_i^{rev}$ to train a local model (denoted as $\tilde{\Theta}_i^t$). Then, the genuine score can be estimated as $\tilde{p}_i^t = 1 - \frac{1}{|\mathcal{D}_i^{rev}|} \sum_{\mathbf{x} \in \mathcal{D}_i^{rev}} \mathbb{I}(G(\mathbf{x} \oplus \delta; \tilde{\Theta}_i^t) = y^{tc})$, where $G(\mathbf{x} \oplus \delta; \tilde{\Theta}_i^t)$ is the predicted label by the global model $\tilde{\Theta}_i^t$ for the trigger-embedded input $\mathbf{x} \oplus \delta$.

**Finding the Optimal $r_i^t$.** The client can use grid search to find $r_i^t$ that achieves the largest $\tilde{p}_i^t + \lambda r_i^t$. After estimating the optimal $r_i^t$, client $i$ can embed the backdoor to $r_i^t$ fraction of training examples and utilize them to perform backdoor attacks based on state-of-the-art methods [3, 54, 59].

**Trigger Optimization.** The attacker can choose an arbitrary static trigger to launch backdoor attacks, or dynamically optimize it to make the attack more effective. Given the fixed location and bounding box of the backdoor trigger, the attacker can optimize the trigger pattern with gradient descent such that it is more likely for a backdoored input to be predicted as the target class. For instance, the attacker can optimize the trigger pattern to minimize the cumulative loss on all training data of malicious clients, i.e., $\delta^t = \operatorname{argmin}_{\delta^*} \sum_{\mathbf{x} \in \cup_{i \in \mathcal{S}_a} \mathcal{D}_i} \ell(\Theta^t(\mathbf{x} \oplus \delta^*), y^{tc})$, where $\Theta^t(\mathbf{x} \oplus \delta^*)$ is the output of the model which is a probability simplex of all possible classes and $\ell$ is a standard loss function (e.g., cross-entropy loss). The attacker can then solve Equation 5 with the optimized trigger. Recall that the attacker aims to find the largest $\tilde{p}_i^t + \lambda r_i^t$. Therefore, if the resulting $\tilde{p}_i^t + \lambda r_i^t$ given $\delta^t$ is lower than that given the previous trigger $\delta^{t-1}$, we let $\delta^t = \delta^{t-1}$.

The complete algorithm for the compromised clients is shown in Algorithm 2 in Appendix C.

## 5 Theoretical Analysis

This section provides a theoretical analysis of FedGame under backdoor attacks. Suppose the global model parameters are in a bounded space. We derive an upper bound for the $L_2$-norm of the difference between the parameters of the global models with and without attacks. To analyze the robustness of FedGame, we make the following assumptions, which are commonly used in the analysis of previous studies [25, 45, 16, 35, 53, 7, 9] on federated learning.

**Assumption 5.1.** The loss function is $\mu$-strongly convex with $L$-Lipschitz continuous gradient. Formally, we have the following for arbitrary $\Theta$ and $\Theta'$:

$$(\nabla_\Theta \ell(\mathbf{z}; \Theta) - \nabla_{\Theta'} \ell(\mathbf{z}; \Theta'))^T (\Theta - \Theta') \geq \mu \|\Theta - \Theta'\|_2^2, \tag{6}$$

$$\|\nabla_\Theta \ell(\mathbf{z}; \Theta) - \nabla_{\Theta'} \ell(\mathbf{z}; \Theta')\|_2 \leq L \|\Theta - \Theta'\|_2, \tag{7}$$

where $\mathbf{z}$ is an arbitrary training example.

**Assumption 5.2.** We assume the gradient $\nabla_\Theta \ell(\mathbf{z}; \Theta)$ is bounded with respect to $L_2$-norm for arbitrary $\Theta$ and $\mathbf{z}$, i.e., there exists some $M \geq 0$ such that

$$\|\nabla_\Theta \ell(\mathbf{z}; \Theta)\|_2 \leq M. \tag{8}$$

Suppose $\Theta_c^t$ is the global model trained by FedGame without any attacks in the $t$th communication round, i.e., each client $i \in \mathcal{S}$ uses its clean local training dataset $\mathcal{D}_i$ to train a local model. Moreover, we assume gradient descent is used by each client to train its local model. Suppose $q_i^t$ is the genuine score for client $i$ without attacks. Moreover, we denote $\beta_i^t = \frac{q_i^t}{\sum_{i \in \mathcal{S}} q_i^t}$ as the normalized genuine score for client $i$. To perform the backdoor attack, we assume a compromised client $i$ can embed the backdoor trigger to $r_i^t$ fraction of training examples in the local training dataset of the client and relabel them as the target class. Those backdoored training examples are used to augment the local training dataset of the client. Suppose $\Theta^t$ is the global model under the backdoor attack in the $t$th communication round with our defense. We denote $\alpha_i^t = \frac{p_i^t}{\sum_{i \in \mathcal{S}} p_i^t}$ as the normalized genuine score for client $i$ with attacks in the $t$th communication round. Formally, we have:

**Lemma 5.3** (Robustness Guarantee for One Communication Round). *Suppose Assumptions 5.1 and 5.2 hold. Moreover, we assume $(1 - r^t)\beta_i^t \leq \alpha_i^t \leq (1 + r^t)\beta_i^t$, where $i \in \mathcal{S}$ and $r^t = \sum_{j \in \mathcal{S}_a} r_j^t$. Then, we have:*

$$\|\Theta^{t+1} - \Theta_c^{t+1}\|_2 \tag{9}$$

$$\leq \sqrt{1 - \eta\mu + 2\eta\gamma^t + \eta^2 L^2 + 2\eta^2 L\gamma^t} \|\Theta^t - \Theta_c^t\|_2 + \sqrt{2\eta\gamma^t(1 + \eta L + 2\eta\gamma^t)} + 2\eta r^t M, \tag{10}$$

*where $\eta$ is the learning rate of the global model, $L$ and $\mu$ are defined in Assumption 5.1, $\gamma^t = \sum_{i \in \mathcal{S}_a} \alpha_i^t r_i^t M$, and $M$ is defined in Assumption 5.2.*

*Proof sketch.* Our idea is to decompose $\|\Theta^{t+1} - \Theta_c^{t+1}\|_2$ into two terms. Then, we derive an upper bound for each term based on the change of the local model updates of clients under backdoor attacks and the properties of the loss function. As a result, our derived upper bound relies on $r_i^t$ for each client $i \in \mathcal{S}_a$, parameters $\mu$, $L$, and $M$ in our assumptions, as well as the parameter difference of the global models in the previous iteration, i.e., $\|\Theta^t - \Theta_c^t\|_2$. Our complete proof is in Appendix B.1. □

In the above lemma, we derive an upper bound of $\left\|\Theta^{t+1} - \Theta_c^{t+1}\right\|_2$ with respect to $\left\|\Theta^t - \Theta_c^t\right\|_2$ for one communication round. In the next theorem, we derive an upper bound of $\left\|\Theta^t - \Theta_c^t\right\|_2$ as $t \to \infty$. We iterative apply Lemma 5.3 for successive values of $t$ and have the following theorem:

**Theorem 5.4** (Robustness Guarantee). *Suppose Assumptions 5.1 and 5.2 hold. Moreover, we assume* $(1 - r^t)\beta_i^t \le \alpha_i^t \le (1 + r^t)\beta_i^t$ *for* $i \in \mathcal{S}$, $\gamma^t \le \gamma$ *and* $r^t \le r$ *hold for all communication round* $t$, *and* $\mu > 2\gamma$, *where* $r^t = \sum_{j \in \mathcal{S}_a} r_j^t$ *and* $\gamma^t = \sum_{i \in \mathcal{S}_a} \alpha_i^t r_i^t M$. *Let the global model learning rate by chosen as* $0 < \eta < \frac{\mu - 2\gamma}{L^2 + 2L\gamma}$. *Then, we have:*

$$\left\|\Theta^t - \Theta_c^t\right\|_2 \le \frac{\sqrt{2\eta\gamma(1 + \eta L + 2\eta\gamma)} + 2\eta r M}{1 - \sqrt{1 - \eta\mu + 2\eta\gamma + \eta^2 L^2 + 2\eta^2 L\gamma}} \tag{11}$$

*holds as* $t \to \infty$.

*Proof sketch.* Given the conditions that $\gamma^t \le \gamma$ and $r^t \le r$ as well as the fact that the right-hand side of Equation 10 is monotonic with respect to $\gamma^t$ and $r^t$, we can replace $\gamma^t$ and $r^t$ in Equation 10 with $\gamma$ and $r$. Then, we iterative apply the equation for successive values of $t$. When $0 < \eta < \frac{\mu - 2\gamma}{L^2 + 2L\gamma}$, we have $0 < 1 - \eta\mu + 2\eta\gamma + \eta^2 L^2 + 2\eta^2 L\gamma < 1$. By letting $t \to \infty$, we can reach the conclusion. The complete proof can be found in Appendix B.2. $\qquad\square$

Our theorem implies that the global model parameters under our defense against adaptive attacks do not deviate too much from those of the global model without attack when the fraction of backdoored training examples $r^t$ is bounded.

## 6 Experiments

In order to thoroughly evaluate the effectiveness of FedGame, we conduct comprehensive experiments on 1) evaluation against three *state-of-the-art* backdoor attacks: Scaling attack [3], DBA [54], and Neurotoxin [59], 2) comparison with six existing baselines including Krum, Median, Norm-Clipping, Differential Privacy, DeepSight, FLTrust, 3) evaluation against strong adaptive attacks, and 4) comprehensive ablation studies.

### 6.1 Experimental Setup

**Datasets and Models.** We use two benchmark datasets: MNIST [23] and CIFAR10 [22] for FL tasks. MNIST has 60,000 training and 10,000 testing images, each of which has a size of $28 \times 28$ belonging to one of 10 classes. CIFAR10 consists of 50,000 training and 10,000 testing images with a size of $32 \times 32$. Each image is categorized into one of 10 classes. For each dataset, we randomly sample 90% of training data for clients, and the remaining 10% of training data is reserved to evaluate our defense when the clean training dataset of the server is from the same domain as those of clients. We use a CNN with two convolution layers (detailed architecture can be found in Table 2 in Appendix) and ResNet-18 [19] which is pre-trained on ImageNet [13] as the global models for MNIST and CIFAR10.

**FL Settings.** We consider two settings: local training data are independently and identically distributed (IID) among clients, and non-IID. We follow the previous work [17] to distribute training data to clients by using a parameter $q$ to control the degree of non-IID, which models the probability that training images from one category are distributed to a particular client (or a set of clients). We set $q = 0.5$ by following [17]. Moreover, we train a global model based on 10 clients for 200 iterations with a global learning rate $\eta = 1.0$. In each communication round, we use SGD to train the local model of each client for two epochs with a local learning rate of 0.01.

**Attack Settings.** We consider three *state-of-the-art* backdoor attacks on federated learning, i.e., Scaling attack [3], DBA [54], and Neurotoxin [59]. For Scaling attack, we set the scaling parameter to be #total clients/($\eta \times$#compromised clients) by following [3]. For Neurotoxin, we set the ratio of masked gradients to be 1%, following the choice in [59]. We use the same backdoor trigger and target class as used in those works. By default, we assume 60% of clients are compromised by an attacker. When the attacker solves the minimax game in Equation 5, we set the default $\lambda = 1$. We explore the

Table 1: Comparison of FedGame with existing defenses under Scaling attack and DBA. The total number of clients is 10, where 60% are compromised. The best results for each setting among FedGame and existing defenses are bold.

| Attacks | Client Data Distribution | Datasets | Metrics | FedAvg (Without attacks) | FedAvg | Krum | Median | Norm-Clipping | DP | Deep-Sight | FLTrust | FedGame In Domain | FedGame Out-of Domain |
|---|---|---|---|---|---|---|---|---|---|---|---|---|---|
| Scaling Attack | IID | MNIST | TA (%) | 99.04 | 98.77 | 98.78 | 99.17 | 95.48 | 92.97 | 97.69 | 97.93 | 98.53 | 98.56 |
| | | | ASR (%) | 9.69 | 99.99 | 99.99 | 99.97 | 98.54 | 99.45 | 20.03 | 16.01 | **9.72** | **9.68** |
| | | CIFAR10 | TA (%) | 81.08 | 80.51 | 76.44 | 80.17 | 80.38 | 43.22 | 76.79 | 75.71 | 74.81 | 74.65 |
| | | | ASR (%) | 8.39 | 99.80 | 99.94 | 99.82 | 99.87 | 99.58 | 98.58 | 99.46 | **8.92** | **9.24** |
| | non-IID | MNIST | TA (%) | 98.98 | 99.15 | 96.88 | 99.12 | 94.54 | 91.52 | 97.39 | 97.68 | 98.28 | 98.34 |
| | | | ASR (%) | 9.73 | 99.99 | 85.03 | 99.98 | 98.16 | 99.54 | 20.03 | 19.61 | **10.42** | **10.88** |
| | | CIFAR10 | TA (%) | 80.25 | 75.35 | 67.66 | 79.54 | 70.18 | 50.79 | 77.76 | 75.08 | 73.88 | 73.57 |
| | | | ASR (%) | 9.67 | 99.92 | 99.92 | 99.99 | 99.63 | 95.01 | 99.03 | 99.82 | **11.76** | **12.03** |
| DBA | IID | MNIST | TA (%) | 99.04 | 99.03 | 98.87 | 98.98 | 98.99 | 98.99 | 98.64 | 97.98 | 97.84 | 98.05 |
| | | | ASR (%) | 9.69 | 100.00 | 10.06 | 99.81 | 99.75 | 99.73 | 15.02 | 10.02 | **9.56** | **9.68** |
| | | CIFAR10 | TA (%) | 81.08 | 80.90 | 76.09 | 80.00 | 80.21 | 41.36 | 72.13 | 75.17 | 73.18 | 72.93 |
| | | | ASR (%) | 8.39 | 93.44 | 94.97 | 91.60 | 91.90 | 86.96 | 83.26 | 66.58 | **8.81** | **9.00** |
| | non-IID | MNIST | TA (%) | 98.98 | 98.98 | 98.58 | 99.13 | 93.98 | 88.91 | 96.65 | 97.62 | 98.58 | 98.59 |
| | | | ASR (%) | 9.73 | 100.00 | 10.52 | 99.85 | 55.97 | 99.66 | 13.21 | 10.19 | **9.97** | **9.83** |
| | | CIFAR10 | TA (%) | 80.25 | 80.15 | 74.31 | 79.78 | 78.78 | 38.17 | 73.83 | 74.57 | 73.52 | 73.21 |
| | | | ASR (%) | 9.67 | 95.03 | 60.06 | 95.00 | 92.51 | 99.51 | 80.75 | 74.35 | **10.62** | **10.67** |

impact of $\lambda$ in our experiments. We randomly sample 10% of the local data of each compromised client as validation data to search for an optimal $r_i^t$. Moreover, we set the granularity of grid search to 0.1 when searching for $r_i^t$.

**Baselines.**   We compare our defense with the following methods: FedAvg [28], Krum [6], Median [56], Norm-Clipping [39], Differential Privacy (DP) [39], DeepSight [36], and FLTrust [7]. Please see Appendix D.2 for parameter settings for those baselines.

**Evaluation Metrics.**   We use *testing accuracy (TA)* and *attack success rate (ASR)* as evaluation metrics. Concretely, TA is the fraction of clean testing inputs that are correctly predicted, and ASR refers to the fraction of backdoored testing inputs that are predicted as the target class.

**Defense Settings.**   We consider two settings: in-domain and out-of-domain. For the in-domain setting, we consider the clean training dataset of the server is from the same domain as the local training datasets of clients. We use the reserved data as the clean training dataset of the server for each dataset. For the out-of-domain setting, we consider the server has a clean training dataset that is from a different domain than the FL task. In particular, we randomly sample 6,000 images from FashionMNIST [52] for MNIST and sample 5,000 images from GTSRB [20] for CIFAR10 as the clean training dataset of the server. Unless otherwise mentioned, we adopt Neural Cleanse [42] to reverse engineer the backdoor trigger and target class.

## 6.2   Experimental Results

We show the results of FedGame compared with existing defenses under IID and non-IID settings in Table 1. We defer the results against Neurotoxin to Appendix D.3 due to space limitations. We have the following observations from the experimental results. First, FedGame outperforms all existing defenses in terms of ASR. In particular, FedGame can reduce ASR to random guessing (i.e., ASR of FedAvg under no attacks) in both IID and non-IID settings for clients as well as both in-domain and out-of-domain settings for the server. Intrinsically, FedGame performs better because our game-theoretic defense enables the defender to optimize its strategy against dynamic, adaptive attacks. We

note that FLTrust outperforms other defenses (except FedGame) in most cases since it exploits a clean training dataset from the same domain as local training datasets of clients. However, FLTrust is not applicable when the server only holds an out-of-domain clean training dataset, while FedGame can relax such an assumption and will still be applicable. Moreover, our experimental results indicate that FedGame achieves comparable performance even if the server holds an out-of-domain clean training dataset.

To further understand our results, we visualize the average genuine (or trust) scores computed by FedGame (or FLTrust) for compromised and benign clients in Appendix D.4. In particular, we find that the genuine scores produced by FedGame are much lower than those produced by FLTrust for compromised clients, which explains why FedGame outperforms FLTrust. Second, FedGame achieves comparable TA with existing defenses, indicating that FedGame preserves the utility of global models.

Furthermore, we show the comparison results of FedGame with existing defenses against the scaling attack when the total number of clients is 30 in Table 4 in Appendix D.5. Our observations are similar, which indicates that FedGame consistently outperforms existing defenses under different numbers of clients and backdoor attacks.

**Impact of $\lambda$.** $\lambda$ is a hyperparameter used by an attacker (see Eqn. 4) when searching for the optimal $r_i^t$ for each compromised client $i$ in each communication round $t$. Figure 1(a) shows the impact of $\lambda$ on ASR of FedGame. The results show that FedGame is insensitive to different $\lambda$'s. The reason is that the genuine score for a compromised client is small when $\lambda$ is large, and the local model of a compromised client is less likely to predict a trigger-embedded input as the target class when $\lambda$ is small. As a result, backdoor attacks with different $\lambda$ are ineffective under FedGame.

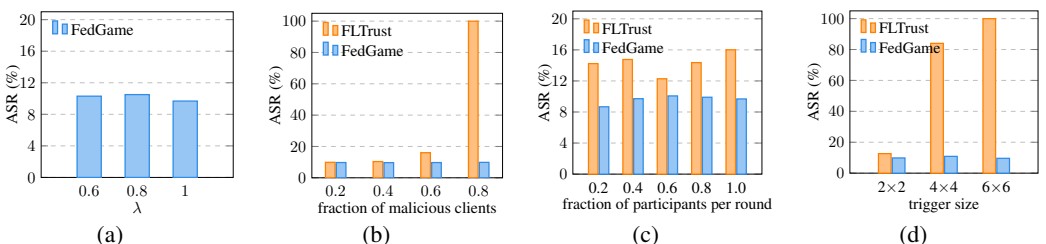

Figure 1: Comparing FedGame and FLTrust under different variations of attack. (a) Different $\lambda$. (b) Different fractions of malicious clients. (c) Different numbers of clients selected in each communication round. (d) Different trigger sizes.

**Impact of the Fraction of Compromised Clients.** Figure 1(b) shows the impact of the fraction of compromised clients on ASR. As the results show, FedGame is effective for a different fraction of compromised clients in both in-domain and out-of-domain settings. In contrast, FLTrust is ineffective when the fraction of compromised clients is large. For instance, FedGame can achieve 9.84% (in-domain) and 10.12% (out-of-domain) ASR even if 80% of clients are compromised on MNIST. Under the same setting, the ASR of FLTrust is 99.95%, indicating that the defense fails.

**Impact of Client Selection.** By default, we consider that all clients are selected in each communication round. We also consider only a subset of clients are selected in each communication round by the server. Figure 1(c) shows our experimental results. Our results suggest that our defense is effective and consistently outperforms FLTrust.

**Computation Cost.** FedGame computes a genuine score for each client in each communication round. Here we demonstrate its computational efficiency. On average, it takes 0.148s to compute a genuine score for each client in each communication round on a single NVIDIA 2080 Ti GPU. We note that the server could from a resourceful tech company (e.g., Google, Meta, Apple), which would have enough computation resources to compute it for millions of clients. Moreover, those local models can be evaluated in parallel.

**Performance under Static Attacks.** In our evaluation, we consider an attacker optimizing the fraction of backdoored training examples. We also evaluate FedGame under existing attacks where the attacker does not optimize it. Under the default setting, FedGame can achieve an ASR of 9.75%, indicating that our defense is effective under static attack.

Furthermore, we discuss the impact of trigger reverse engineering methods, the total number of clients, and the size of clean data of the server in Appendix D.6.

### 6.3 Adaptive Attacks

In this subsection, we discuss some potential adaptive strategies that may be leveraged by the attacker to evade our defense.

**Data Replacing Attack.** By default, we consider an attacker optimizing the fraction of backdoored training examples *added* to the local training dataset of a compromised client to maximize backdoor effectiveness. We also consider an attacker who *replaces* a fraction of the local training data with backdoored samples and optimizes such fraction. Under our default setting, the ASR of FedGame is 9.71% and 10.69% on MNIST and CIFAR10, respectively, indicating that our defense is still effective under data replacing attacks.

**Variation in Triggers.** We demonstrate that our defense is robust to variation in triggers. In particular. we try triggers with sizes $2 \times 2$, $4 \times 4$, and $6 \times 6$ under the default setting. Figure 1(d) compares FedGame with FLTrust. The results demonstrate that FedGame is consistently effective under triggers with different sizes.

We note that although an attacker can slightly manipulate the parameters of local models such that they are more similar to those of benign clients, FedGame does not rely on model parameters for detection. Instead, FedGame leverages the model behaviors, i.e., whether the model predicts inputs with our reverse-engineered trigger as the target class. As a result, our defense would still be effective even if the change in the model parameters is small as long as the model has backdoor behavior (this is required to make the attack effective). This is also the reason why our defense is better than existing methods such as FLTrust which leverages model parameters for defense.

## 7 Conclusion and Future Work

In this work, we propose FedGame, a general game-theory based defense against adaptive backdoor attacks in federated learning. Our formulated minimax game enables the defender and attacker to dynamically optimize their strategies. Moreover, we respectively design solutions for both of them to solve the minimax game. We perform theoretical analysis and empirical evaluations for our framework. Our results demonstrate the effectiveness of FedGame under strategic backdoor attackers. Moreover, FedGame achieves significantly higher robustness than baselines in different settings.

We consider that an attacker could optimize the poisoning ratio and trigger pattern in our work. We believe it is an interesting future work to consider other factors for the attacker (e.g., trigger size, consideration of long-term goal, design of new loss functions [12], poisoned neurons selection [2]). Moreover, we consider a zero-sum Stackelberg game in this work. Another interesting future work is to consider other game formulations, e.g., Bayesian games.

## Acknowledgements

This work is partially supported by the National Science Foundation (NSF) under grants No.1910100, No.2046726, No.2153136, No.2229876, DARPA GARD, the National Aeronautics and Space Administration (NASA) under grant No.80NSSC20M0229, Alfred P. Sloan Fellowship, Office of Naval Research (ONR) under grant N00014-23-1-2386, Air Force Office of Scientific Research (AFOSR) under grant FA9550-23-1-0208, and the Amazon research award.

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

## A   Interpretation of the Minimax Objective

We note that the ultimate goal of the defender is to obtain a clean model while the attacker wants the global model to be poisoned. In other words, the server (or attacker) wishes the attack success rate of the global model to be small (large). It is very challenging to achieve this objective directly because 1) the server does not know the exact trigger and target class used by the attacker, and 2) the attacker needs to adapt its attack strategy based on the server's strategy. To address the challenge, we consider an alternative goal (i.e., our objective in Equation 5, where we wish to assign small weights to compromised clients when performing a weighted average to update the global model. Our idea is that when the weights for compromised clients are very small, our global model is less likely to be affected by the attack. As a result, the attacker's strategy is to maximize the genuine scores for compromised clients while ensuring their local models are backdoored to maximize the attack effectiveness for the global model. Thus, our objective in Equation 5 could translate to the ultimate goal. Our empirical results show the effectiveness of our defense.

## B   Complete Proofs

### B.1   Proof of Lemma 5.3

We first present some preliminary lemmas that will be invoked for proving Lemma 5.3.

**Lemma B.1.** *Suppose $\mathcal{D}_i$ is the clean local training dataset of the client $i$. An attacker can inject the backdoor trigger to $r_i^t$ fraction of training examples in $\mathcal{D}_i$ and relabel them as the target class. We use $\mathcal{D}_i'$ to denote the set of backdoored training examples where $r_i^t = \frac{|\mathcal{D}_i'|}{|\mathcal{D}_i|}$. Given two arbitrary $\Theta$ and $\Theta_c$, we let $g_i = \frac{1}{|\mathcal{D}_i \cup \mathcal{D}_i'|}\nabla_\Theta \sum_{\mathbf{z}\in\mathcal{D}_i\cup\mathcal{D}_i'} \ell(\mathbf{z};\Theta)$ and $h_i = \frac{1}{|\mathcal{D}_i|}\nabla_{\Theta_c} \sum_{\mathbf{z}\in\mathcal{D}_i} \ell(\mathbf{z};\Theta_c)$. We then have that*

$$(\Theta - \Theta_c)^T(g_i - h_i) \geq (0.5\mu - r_i^t M)\|\Theta - \Theta_c\|_2^2 - r_i^t M, \tag{12}$$

$$\|g_i - h_i\|_2 \leq L\|\Theta - \Theta_c\|_2 + 2r_i^t M. \tag{13}$$

*Proof.* We first prove Equation 12. We have the following relations:

$$(\Theta - \Theta_c)^T(g_i - h_i)$$

$$=(\Theta - \Theta_c)^T(\frac{1}{|\mathcal{D}_i \cup \mathcal{D}_i'|}\sum_{\mathbf{z}'\in\mathcal{D}_i\cup\mathcal{D}_i'}\nabla_\Theta\ell(\mathbf{z}';\Theta) - \frac{1}{|\mathcal{D}_i|}\sum_{\mathbf{z}\in\mathcal{D}_i}\nabla_{\Theta_c}\ell(\mathbf{z};\Theta_c)) \quad \triangleright \text{definition of } g_i \text{ and } h_i$$
$$\tag{14}$$

$$=(\Theta - \Theta_c)^T(\frac{1}{(1+r_i^t)|\mathcal{D}_i|}\sum_{\mathbf{z}'\in\mathcal{D}_i\cup\mathcal{D}_i'}\nabla_\Theta\ell(\mathbf{z}';\Theta) - \frac{1}{|\mathcal{D}_i|}\sum_{\mathbf{z}\in\mathcal{D}_i}\nabla_{\Theta_c}\ell(\mathbf{z};\Theta_c)) \quad \triangleright r_i^t = \frac{|\mathcal{D}_i'|}{|\mathcal{D}_i|} \tag{15}$$

$$=\frac{1}{|\mathcal{D}_i|(1+r_i^t)}(\Theta - \Theta_c)^T(\sum_{\mathbf{z}'\in\mathcal{D}_i\cup\mathcal{D}_i'}\nabla_\Theta\ell(\mathbf{z}';\Theta) - (1+r_i^t)\sum_{\mathbf{z}\in\mathcal{D}_i}\nabla_{\Theta_c}\ell(\mathbf{z};\Theta_c)) \tag{16}$$

$$=\frac{1}{|\mathcal{D}_i|(1+r_i^t)}(\Theta - \Theta_c)^T(\sum_{\mathbf{z}'\in\mathcal{D}_i}\nabla_\Theta\ell(\mathbf{z}';\Theta) - \sum_{\mathbf{z}\in\mathcal{D}_i}\nabla_{\Theta_c}\ell(\mathbf{z};\Theta_c)$$

$$+ \sum_{\mathbf{z}'\in\mathcal{D}_i'}\nabla_\Theta\ell(\mathbf{z}';\Theta) - r_i^t\sum_{\mathbf{z}\in\mathcal{D}_i}\nabla_{\Theta_c}\ell(\mathbf{z};\Theta_c)) \tag{17}$$

$$=\frac{1}{|\mathcal{D}_i|(1+r_i^t)}(\sum_{\mathbf{z}\in\mathcal{D}_i}(\Theta - \Theta_c)^T(\nabla_\Theta\ell(\mathbf{z};\Theta) - \nabla_{\Theta_c}\ell(\mathbf{z};\Theta_c))$$

$$+ (\Theta - \Theta_c)^T(\sum_{\mathbf{z}'\in\mathcal{D}_i'}\nabla_\Theta\ell(\mathbf{z}';\Theta) - r_i^t\sum_{\mathbf{z}\in\mathcal{D}_i}\nabla_{\Theta_c}\ell(\mathbf{z};\Theta_c))) \tag{18}$$

$$\geq\frac{1}{|\mathcal{D}_i|(1+r_i^t)}(\sum_{\mathbf{z}\in\mathcal{D}_i}(\Theta - \Theta_c)^T(\nabla_\Theta\ell(\mathbf{z};\Theta) - \nabla_{\Theta_c}\ell(\mathbf{z};\Theta_c))$$

$$- \|(\Theta - \Theta_c)^T(\sum_{\mathbf{z}'\in\mathcal{D}_i'}\nabla_\Theta\ell(\mathbf{z}';\Theta) - r_i^t\sum_{\mathbf{z}\in\mathcal{D}_i}\nabla_{\Theta_c}\ell(\mathbf{z};\Theta_c))\|_1) \quad \triangleright \forall x, x \geq -\|x\|_1 \tag{19}$$

$$\geq \frac{1}{|\mathcal{D}_i|(1+r_i^t)}(\sum_{\mathbf{z}\in\mathcal{D}_i}(\Theta-\Theta_c)^T(\nabla_\Theta\ell(\mathbf{z};\Theta)-\nabla_{\Theta_c}\ell(\mathbf{z};\Theta_c))$$

$$-\|\Theta-\Theta_c\|_2\cdot\|\sum_{\mathbf{z}'\in\mathcal{D}_i'}\nabla_\Theta\ell(\mathbf{z}';\Theta)-r_i^t\sum_{\mathbf{z}\in\mathcal{D}_i}\nabla_{\Theta_c}\ell(\mathbf{z};\Theta_c)\|_2) \quad \triangleright \text{ Cauchy–Schwarz inequality}$$

$$\geq\frac{1}{|\mathcal{D}_i|(1+r_i^t)}(\sum_{\mathbf{z}\in\mathcal{D}_i}(\Theta-\Theta_c)^T(\nabla_\Theta\ell(\mathbf{z};\Theta)-\nabla_{\Theta_c}\ell(\mathbf{z};\Theta_c))$$

$$-\|\Theta-\Theta_c\|_2\cdot(\sum_{\mathbf{z}'\in\mathcal{D}_i'}\|\nabla_\Theta\ell(\mathbf{z}';\Theta)\|_2+r_i^t\sum_{\mathbf{z}\in\mathcal{D}_i}\|\nabla_{\Theta_c}\ell(\mathbf{z};\Theta_c)\|_2) \quad \triangleright \text{ triangle inequality}$$

$$\geq\frac{1}{|\mathcal{D}_i|(1+r_i^t)}(\mu|\mathcal{D}_i|\,\|\Theta-\Theta_c\|_2^2-2r_i^t|\mathcal{D}_i|M\,\|\Theta-\Theta_c\|_2) \quad \triangleright \text{ Assumption 5.1 and 5.2} \tag{20}$$

$$=\frac{\mu}{1+r_i^t}\|\Theta-\Theta_c\|_2^2-\frac{1}{1+r_i^t}2r_i^tM\,\|\Theta-\Theta_c\|_2) \tag{21}$$

$$\geq 0.5\mu\,\|\Theta-\Theta_c\|_2^2-2r_i^tM\,\|\Theta-\Theta_c\|_2 \quad \triangleright r_i^t\in[0,1] \tag{22}$$

$$\geq 0.5\mu\,\|\Theta-\Theta_c\|_2^2-r_i^tM\,\|\Theta-\Theta_c\|_2^2-r_i^tM) \tag{23}$$

$$=(0.5\mu-r_i^tM)\,\|\Theta-\Theta_c\|_2^2-r_i^tM, \tag{24}$$

where Equation 23 holds based on the fact that $-2r_i^tM\,\|\Theta-\Theta_c\|_2\geq-r_i^tM\,\|\Theta-\Theta_c\|_2^2-r_i^tM$ for $\forall r_i^t\geq 0$ and $\forall M\geq 0$.

In the following, we prove inequality 13. We have that

$$\|g_i-h_i\|_2$$

$$=\frac{1}{|\mathcal{D}_i|(1+r_i^t)}\|\sum_{\mathbf{z}'\in\mathcal{D}_i\cup\mathcal{D}_i'}\nabla_\Theta\ell(\mathbf{z}';\Theta)-(1+r_i^t)\sum_{\mathbf{z}\in\mathcal{D}_i}\nabla_{\Theta_c}\ell(\mathbf{z};\Theta_c)\|_2 \quad \triangleright \text{ definition of } g_i \text{ and } h_i$$

$$\tag{25}$$

$$=\frac{1}{|\mathcal{D}_i|(1+r_i^t)}\|\sum_{\mathbf{z}'\in\mathcal{D}_i'}\nabla_\Theta\ell(\mathbf{z}';\Theta)+\sum_{\mathbf{z}'\in\mathcal{D}_i}\nabla_\Theta\ell(\mathbf{z}';\Theta)-(1+r_i^t)\sum_{\mathbf{z}\in\mathcal{D}_i}\nabla_{\Theta_c}\ell(\mathbf{z};\Theta_c)\|_2 \tag{26}$$

$$\leq\frac{1}{|\mathcal{D}_i|(1+r_i^t)}\|\sum_{\mathbf{z}'\in\mathcal{D}_i'}\nabla_\Theta\ell(\mathbf{z}';\Theta)-r_i^t\sum_{\mathbf{z}\in\mathcal{D}_i}\nabla_{\Theta_c}\ell(\mathbf{z};\Theta_c)\|_2$$

$$+\frac{1}{|\mathcal{D}_i|(1+r_i^t)}\|\sum_{\mathbf{z}'\in\mathcal{D}_i}\nabla_\Theta\ell(\mathbf{z}';\Theta)-\sum_{\mathbf{z}\in\mathcal{D}_i}\nabla_{\Theta_c}\ell(\mathbf{z};\Theta_c)\|_2 \quad \triangleright \text{ triangle inequality} \tag{27}$$

$$\leq\frac{1}{1+r_i^t}(2r_i^tM+L\|\Theta-\Theta_c\|_2) \tag{28}$$

$$\leq 2r_i^tM+L\|\Theta-\Theta_c\|_2 \quad \triangleright r_i^t\in[0,1] \tag{29}$$

where Equation 28 is due to Assumption 5.1 and 5.2. $\qquad\square$

Given Lemma B.1, we prove Lemma 5.3 as follows. Recall that we have $\alpha_i^t=\frac{p_i^t}{\sum_{i\in S}p_i^t}$ and $\beta_i^t=\frac{q_i^t}{\sum_{i\in S}q_i^t}$.

$$\|\Theta^{t+1}-\Theta_c^{t+1}\|_2 \tag{30}$$

$$=\|\Theta^t-\eta\sum_{i\in\mathcal{S}}\alpha_i^tg_i^t-(\Theta_c^t-\eta\sum_{i\in\mathcal{S}}\beta_i^th_i^t)\|_2 \quad \triangleright \text{ gradient descent for } \Theta^{t+1} \text{ and } \Theta_c^{t+1} \tag{31}$$

$$=\|\Theta^t-\eta\sum_{i\in\mathcal{S}}\alpha_i^tg_i^t-(\Theta_c^t-\eta\sum_{i\in\mathcal{S}}(\alpha_i^t+\beta_i^t-\alpha_i^t)h_i^t)\|_2 \tag{32}$$

$$=\|\Theta^t-\Theta_c^t-\eta\sum_{i\in\mathcal{S}}\alpha_i^t(g_i^t-h_i^t)+(\eta\sum_{i\in\mathcal{S}}(\beta_i^t-\alpha_i^t)h_i^t)\|_2 \quad \triangleright \text{ rearranging Equation 32} \tag{33}$$

$$\leq \|\Theta^t - \Theta_c^t - \eta \sum_{i \in \mathcal{S}} \alpha_i^t (g_i^t - h_i^t)\|_2 + \|\eta \sum_{i \in \mathcal{S}} (\beta_i^t - \alpha_i^t) h_i^t\|_2. \quad \triangleright \text{triangle inequality} \tag{34}$$

Next, we respectively derive an upper bound for the first and second terms in Equation 34. To derive the upper bound for the first term, we have that

$$\|\Theta^t - \Theta_c^t - \eta \sum_{i \in \mathcal{S}} \alpha_i^t (g_i^t - h_i^t)\|_2^2$$

$$= \|\Theta^t - \Theta_c^t\|_2^2 - 2\eta (\Theta^t - \Theta_c^t)^T (\sum_{i \in \mathcal{S}} \alpha_i^t (g_i^t - h_i^t)) + \eta^2 \|\sum_{i \in \mathcal{S}} \alpha_i^t (g_i^t - h_i^t)\|_2^2 \tag{35}$$

$$= S_1 + S_2 + S_3, \tag{36}$$

where $S_1 = \|\Theta^t - \Theta_c^t\|_2^2$, $S_2 = -2\eta (\Theta^t - \Theta_c^t)^T (\sum_{i \in \mathcal{S}} \alpha_i^t (g_i^t - h_i^t))$, and $S_3 = \eta^2 \|\sum_{i \in \mathcal{S}} \alpha_i^t (g_i^t - h_i^t)\|_2^2$. Next, we will bound $S_2$ and $S_3$. We denote $\gamma^t = \sum_{i \in \mathcal{S}_a} \alpha_i^t r_i^t M$. Note that we have $\gamma^t = \sum_{i \in \mathcal{S}} \alpha_i^t r_i^t M$ since $r_i^t = 0$ for $\forall i \in \mathcal{S} \setminus \mathcal{S}_a$. We bound $S_2$ as follows.

$$S_2$$

$$= -2\eta (\Theta^t - \Theta_c^t)^T (\sum_{i \in \mathcal{S}} \alpha_i^t (g_i^t - h_i^t)) \tag{37}$$

$$= -2\eta \sum_{i \in \mathcal{S}} \alpha_i^t (\Theta^t - \Theta_c^t)^T (g_i^t - h_i^t) \tag{38}$$

$$\leq -2\eta \sum_{i \in \mathcal{S}} \alpha_i^t ((0.5\mu - r_i^t M) \|\Theta^t - \Theta_c^t\|_2^2 - r_i^t M) \tag{39}$$

$$= -2\eta ((0.5\mu - \sum_{i \in \mathcal{S}} \alpha_i^t r_i^t M) \|\Theta^t - \Theta_c^t\|_2^2 - \sum_{i \in \mathcal{S}_a} \alpha_i^t r_i^t M) \tag{40}$$

$$= (-\eta\mu + 2\eta\gamma^t) \|\Theta^t - \Theta_c^t\|_2^2 + 2\eta\gamma^t, \quad \triangleright \text{definition of } \gamma^t \tag{41}$$

where inequality 39 holds by Lemma B.1 and the fact that $\eta, \alpha_i^t \geq 0$. We bound $S_3$ as follows.

$$S_3$$

$$= \eta^2 \|\sum_{i \in \mathcal{S}} \alpha_i^t (g_i^t - h_i^t)\|_2^2 \tag{42}$$

$$\leq \eta^2 (\sum_{i \in \mathcal{S}} \alpha_i^t \|(g_i^t - h_i^t)\|_2)^2 \tag{43}$$

$$\leq \eta^2 (\sum_{i \in \mathcal{S}} \alpha_i^t (2r_i^t M + L\|\Theta - \Theta_c\|_2)^2 \quad \triangleright \text{Lemma B.1} \tag{44}$$

$$= \eta^2 (2\gamma^t + L\|\Theta - \Theta_c\|_2)^2 \tag{45}$$

$$= \eta^2 (L^2 \|\Theta - \Theta_c\|_2^2 + 4\gamma^t L \|\Theta - \Theta_c\|_2 + 4[\gamma^t]^2) \tag{46}$$

$$\leq \eta^2 (L^2 \|\Theta - \Theta_c\|_2^2 + 2\gamma^t L \|\Theta - \Theta_c\|_2^2 + 2L\gamma^t + 4[\gamma^t]^2) \tag{47}$$

$$= \eta^2 \cdot ((L^2 + 2L\gamma^t) \cdot \|\Theta - \Theta_c\|_2^2 + 2L\gamma^t + 4[\gamma^t]^2) \tag{48}$$

where Equation 47 is based on the fact that $4\gamma^t L \|\Theta - \Theta_c\|_2 \leq 2\gamma^t L \|\Theta - \Theta_c\|_2^2 + 2\gamma^t L$ when $\gamma^t L \geq 0$.

Given the upper bounds of $S_2$ and $S_3$, we can bound $\|\Theta^t - \Theta_c^t - \eta \sum_{i \in \mathcal{S}} \alpha_i^t (g_i^t - h_i^t)\|_2^2$ as follows.

$$\|\Theta^t - \Theta_c^t - \eta \sum_{i \in \mathcal{S}} \alpha_i^t (g_i^t - h_i^t)\|_2^2 \tag{49}$$

$$= S_1 + S_2 + S_3 \tag{50}$$

$$\leq \|\Theta - \Theta_c\|_2^2 + (-\eta\mu + 2\eta\gamma^t) \|\Theta^t - \Theta_c^t\|_2^2 + 2\eta\gamma^t$$

$$+ (\eta^2 L^2 + \eta^2 2L\gamma^t)\left\|\Theta^t - \Theta_c^t\right\|_2^2 + \eta^2 2L\gamma^t + \eta^2 4[\gamma^t]^2 \tag{51}$$

$$= (1 - \eta\mu + 2\eta\gamma^t + \eta^2 L^2 + 2\eta^2 L\gamma^t)\left\|\Theta^t - \Theta_c^t\right\|_2^2 + 2\eta\gamma^t + 2\eta^2 L\gamma^t + 4\eta^2[\gamma^t]^2 \tag{52}$$

Next, we will derive an upper bound for $\left\|\eta\sum_{i\in\mathcal{S}}(\beta_i^t - \alpha_i^t)h_i^t\right\|_2$. We denote $r^t = \sum_{i\in\mathcal{S}_a} r_i^t$. Note that we have that $r^t = \sum_{i\in\mathcal{S}} r_i^t$ also holds since $r_i^t = 0$ for $\forall i \in \mathcal{S} \setminus \mathcal{S}_a$. Given the assumption that $(1 - r^t)\alpha_i^t \le \beta_i^t \le (1 + r^t)\alpha_i^t$, we have

$$\|\eta\sum_{i\in\mathcal{S}}(\beta_i^t - \alpha_i^t)h_i^t\|_2 \le \eta\sum_{i\in\mathcal{S}}|\beta_i^t - \alpha_i^t|\left\|h_i^t\right\|_2 \le 2\eta r^t M, \tag{53}$$

where the first inequality is due to triangle inequality and the second inequality is based on the assumption that $\|h_i^t\|_2 \le M$. Therefore, we have:

$$\|\Theta^{(t+1)} - \Theta_c^{(t+1)}\|_2$$

$$\le \|\Theta^t - \Theta_c^t - \eta\sum_{i\in\mathcal{S}}\alpha_i^t(g_i^t - h_i^t)\|_2^2 + \|\eta\sum_{i\in\mathcal{S}}(\beta_i^t - \alpha_i^t)h_i^t\|_2 \quad \triangleright \text{Equation 30, 34} \tag{54}$$

$$\le \sqrt{(1 - \eta\mu + 2\eta\gamma^t + \eta^2 L^2 + 2\eta^2 L\gamma^t)\left\|\Theta^t - \Theta_c^t\right\|_2^2 + 2\eta\gamma^t(1 + \eta L + 2\eta\gamma^t)} \tag{55}$$

$$+ 2\eta r^t M \quad \triangleright \text{Equation 49, 52, 53} \tag{56}$$

$$\le \sqrt{1 - \eta\mu + 2\eta\gamma^t + \eta^2 L^2 + 2\eta^2 L\gamma^t}\left\|\Theta^t - \Theta_c^t\right\|_2 + \sqrt{2\eta\gamma^t(1 + \eta L + 2\eta\gamma^t)} + 2\eta r^t M, \tag{57}$$

where the last inequality holds due to the fact that $\sqrt{a+b} \le \sqrt{a} + \sqrt{b}$ for $\forall a \ge 0$ and $\forall b \ge 0$, which completes our proof for Lemma 5.3.

## B.2 Proof of Theorem 5.4

We denote $A_t = \sqrt{1 - \eta\mu + 2\eta\gamma^t + \eta^2 L^2 + 2\eta^2 L\gamma^t}$, $A = \sqrt{1 - \eta\mu + 2\eta\gamma + \eta^2 L^2 + 2\eta^2 L\gamma}$, $B_t = \sqrt{2\eta\gamma^t(1 + \eta L + 2\eta\gamma^t)} + 2\eta r^t M$, and $B = \sqrt{2\eta\gamma(1 + \eta L + 2\eta\gamma)} + 2\eta r M$. Since $\gamma^t \le \gamma$ and $r^t \le r$, we have $A_t \le A$ and $B_t \le B$. Thus, based on Lemma 5.3, we have:

$$\left\|\Theta^t - \Theta_c^t\right\|_2 \le A\left\|\Theta^{t-1} - \Theta_c^{t-1}\right\|_2 + B. \tag{58}$$

Then, we can iteratively apply the above equation to prove our theorem. In particular, we have:

$$\left\|\Theta^t - \Theta_c^t\right\|_2$$

$$\le A\left\|\Theta^{t-1} - \Theta_c^{t-1}\right\|_2 + B \tag{59}$$

$$\le A(A\left\|\Theta^{t-2} - \Theta_c^{t-2}\right\|_2 + B) + B \tag{60}$$

$$= A^2\left\|\Theta^{t-2} - \Theta_c^{t-2}\right\|_2 + (A^1 + A^0)B \tag{61}$$

$$\le A^t\left\|\Theta^0 - \Theta_c^0\right\|_2 + (A^{t-1} + A^{t-2} + \cdots + A^0)B \tag{62}$$

$$= A^t\left\|\Theta^0 - \Theta_c^0\right\|_2 + \frac{1 - A^t}{1 - A}B \tag{63}$$

$$= (\sqrt{1 - \eta\mu + 2\eta\gamma + \eta^2 L^2 + 2\eta^2 L\gamma})^t\left\|\Theta^0 - \Theta_c^0\right\|_2$$

$$+ \frac{1 - (\sqrt{1 - \eta\mu + 2\eta\gamma + \eta^2 L^2 + 2\eta^2 L\gamma})^t}{1 - \sqrt{1 - \eta\mu + 2\eta\gamma + \eta^2 L^2 + 2\eta^2 L\gamma}}(\sqrt{2\eta\gamma(1 + \eta L + 2\eta\gamma)} + 2\eta r M), \tag{64}$$

When the learning rate satisfies $0 < \eta < \frac{\mu - 2\gamma}{L^2 + 2L\gamma}$, we have that $0 < 1 - \eta\mu + 2\eta\gamma + \eta^2 L^2 + 2\eta^2 L\gamma < 1$. Therefore, the upper bound becomes $\frac{\sqrt{2\eta\gamma(1+\eta L+2\eta\gamma)}+2\eta r M}{1-\sqrt{1-\eta\mu+2\eta\gamma+\eta^2 L^2+2\eta^2 L\gamma}}$ as $t \to \infty$. Hence, we prove our Theorem 5.4.

## C Complete Algorithms

### C.1 Complete Algorithm of FedGame

Algorithm 1 shows the complete algorithm of FedGame. In Line 3, we construct an auxiliary global model. In Line 4, the function REVERSEENGINEER is used to reverse engineer the backdoor trigger

and target class. In Line 6, we compute the local model of client $i$ based on its local model update. In Line 7, we compute a genuine score for client $i$. In Line 9, we update the global model based on genuine scores and local model updates of clients.

---

**Algorithm 1** FEDGAME

---

**Input:** $\Theta^t$ (global model in the $t^{\text{th}}$ communication round), $g_i^t, i \in \mathcal{S}$ (local model updates of clients), $\mathcal{D}_s$ (clean training dataset of server), $\eta$ (learning rate of global model).
**Output:** $\Theta^{t+1}$ (global model for the $(t+1)^{\text{th}}$ communication round)
$\Theta_a^t = \Theta^t + \frac{1}{|\mathcal{S}|} \sum_{i \in \mathcal{S}} g_i^t$
$\delta_{re}, y_{re}^{tc} = \text{REVERSEENGINEER}(\Theta_a^t)$
**for** $i \in \mathcal{S}$ **do**
    $\Theta_i^t = \Theta^t + g_i^t$
    $p_i^t = 1 - \frac{1}{|\mathcal{D}_s|} \sum_{\mathbf{x} \in \mathcal{D}_s} \mathbb{I}(G(\mathbf{x} \oplus \delta_{re}; \Theta_i^t) = y_{re}^{tc})$
**end for**
$\Theta^{t+1} = \Theta^t + \eta \frac{1}{\sum_{i \in \mathcal{S}} p_i^t} \sum_{i \in \mathcal{S}} p_i^t g_i^t$
**return** $\Theta^{t+1}$

---

### C.2 Complete Algorithm for a Compromised Client

Algorithm 2 shows the complete algorithm for a compromised client. In Line 3, we randomly subsample $\rho_i$ fraction of training data from $\mathcal{D}_i$. In Line 7, the function CREATEBACKDOOREDDATA is used to generate backdoored training examples by embedding the backdoor trigger $\delta$ to $\lfloor \min(j * \zeta, 1) | \mathcal{D}_i \setminus \mathcal{D}_i^{rev} | \rfloor$ training examples in $\mathcal{D}_i \setminus \mathcal{D}_i^{rev}$ and relabel them as $y^{tc}$, where $|\cdot|$ measures the number of elements in a set. In Line 8, the function TRAININGLOCALMODEL is used to train the local model on the training dataset $\mathcal{D}_i' \cup \mathcal{D}_i \setminus \mathcal{D}_i^{rev}$. In Line 9, we estimate a genuine score. In Line 15, we use the function CREATEBACKDOOREDDATA to generate backdoored training examples by embedding the backdoor trigger $\delta$ to $\lfloor \min(o * \zeta, 1) | \mathcal{D}_i | \rfloor$ training examples in $\mathcal{D}_i$ and relabel them as $y^{tc}$. In Line 16, we use the function TRAININGLOCALMODEL to train a local model and utilize existing state-of-the-art attacks to inject the backdoor based on the training dataset $\mathcal{D}_i' \cup \mathcal{D}_i$.

---

**Algorithm 2** ALGORITHM FOR A COMPROMISED CLIENT

---

1: **Input:** $\Theta^t$ (global model in the $t^{\text{th}}$ communication round), $\mathcal{D}_i$ (local training dataset of client $i$), $\rho_i$ (fraction of reserved data to find optimal $r_i^t$), $\zeta$ (granularity of searching for $r_i^t$), $\delta$ (backdoor trigger), $y^{tc}$ (target class), and $\lambda$ (hyperparameter)
2: **Output:** $g_i^t$ (local model update)
3: $\mathcal{D}_i^{rev} = \text{RANDOMSAMPLING}(\mathcal{D}_i, \rho_i)$
4: $count = \lceil \frac{1}{\zeta} \rceil$
5: $max\_value, o \leftarrow 0, 0$
6: **for** $j \leftarrow 0$ to count **do**
7:     $\mathcal{D}_i' = \text{CREATEBACKDOOREDDATA}(\mathcal{D}_i \setminus \mathcal{D}_i^{rev}, \delta, y^{tc}, \min(j * \zeta, 1))$
8:     $\Theta_{ij} = \text{TRAININGLOCALMODEL}(\Theta^t, \mathcal{D}_i' \cup \mathcal{D}_i \setminus \mathcal{D}_i^{rev})$
9:     $p_{ij} = 1 - \frac{1}{|\mathcal{D}_i^{rev}|} \sum_{\mathbf{x} \in \mathcal{D}_i^{rev}} \mathbb{I}(G(\mathbf{x} \oplus \delta; \Theta_{ij}) = y^{tc})$
10:     **if** $p_{ij} + \lambda \min(j * \zeta, 1) > max\_value$ **then**
11:         $o = j$
12:         $max\_value = p_{ij} + \lambda \min(j * \zeta, 1)$
13:     **end if**
14: **end for**
15: $\mathcal{D}_i' = \text{CREATEBACKDOOREDDATA}(\mathcal{D}_i, \delta, y^{tc}, \min(o * \zeta, 1))$
16: $\Theta_i^t = \text{TRAININGLOCALMODEL}(\Theta^t, \mathcal{D}_i' \cup \mathcal{D}_i)$
17: **return** $\Theta_i^t - \Theta^t$

---

# D    Additional Experimental Setup and Results

## D.1    Architecture of Global Model

Table 2 shows the global model architecture on MNIST dataset.

## D.2    Parameter Setting for Compared Baselines

Recall that we compare our defense with the following methods: FedAvg [28], Krum [6], Median [56], Norm-Clipping [39], Differential Privacy (DP) [39], DeepSight [36], and FLTrust [7]. FedAvg is non-robust while Krum and Median are two Byzantine-robust baselines. Norm-Clipping clips the $L_2$-norm of local model updates to a given threshold $\mathcal{T}_N$. We set $\mathcal{T}_N = 0.01$ for MNIST and $\mathcal{T}_N = 0.1$ for CIFAR10. DP first clips the $L_2$-norm of a local model update to a threshold $\mathcal{T}_D$ and then adds Gaussian noise. We set $\mathcal{T}_D = 0.05$ for MNIST and $\mathcal{T}_D = 0.5$ for CIFAR10. We set the standard deviation of noise to be $0.01$ for both datasets. In FLTrust, the server uses its clean dataset to compute a server model update and assigns a trust score to each client by leveraging the similarity between the server model update and the local model update. We set the clean training dataset of the server to be the same as FedGame in our comparison. Note that FLTrust is not applicable when the clean training dataset of the server is from a different domain from those of clients.

Table 2: Architecture of the convolutional neural network for MNIST.

| Type | Parameters |
|---|---|
| Convolution | $3 \times 3$, stride=1, 16 kernels |
| Activation | ReLU |
| Max Pooling | $2 \times 2$ |
| Convolution | $4 \times 4$, stride=2, 32 kernels |
| Activation | ReLU |
| Max Pooling | $2 \times 2$ |
| Fully Connected | $800 \times 500$ |
| Activation | ReLU |
| Fully Connected | $500 \times 10$ |

## D.3    Performance of FedGame against Neurotoxin

In Table 3, we compare our FedGame with other defense baselines against Neurotoxin [59]. We can observe that our FedGame is consistently more effective than existing defenses. Our observation is consistent with the experimental results for Scaling attack and DBA attack in Table 1.

Table 3: Comparison of FedGame with existing defenses against Neurotoxin on MNIST under IID setting. The total number of clients is 10 with 60% compromised. The best results when respectively comparing FedGame in each setting with existing defenses are bold.

| Metrics | FedAvg (No attacks) | Defenses (Under attacks) | | | | | | | FedGame | |
|---|---|---|---|---|---|---|---|---|---|---|
| | | FedAvg | Krum | Median | Norm-Clipping | DP | Deep-Sight | FLTrust | In-domain | Out-of-domain |
| TA (%) | 99.04 | 99.02 | 99.32 | 99.08 | 90.75 | 95.28 | 96.36 | 95.73 | 97.27 | 97.33 |
| ASR (%) | 9.69 | 99.97 | 99.98 | 99.99 | 99.36 | 99.27 | 89.02 | 13.02 | **9.93** | **10.03** |

## D.4    Visualization of Genuine Score of FedGame and Trust Score of FLTrust

Our FedGame computes a genuine score for each client which quantifies the extent to which a client is benign in each communication round. Intuitively, our FedGame would be effective if the genuine score is small for a compromised client but is large for a benign one. FLTrust [7] computes a trust score for each client in each communication round. Similarly, FLTrust would be effective if the trust score is small for a compromised client but large for a benign one. Figure 2 visualizes the

average genuine or trust scores for compromised and benign clients of FedGame and FLTrust on the MNIST dataset. We have the following observations from the figures. First, the average genuine score computed by FedGame drops to 0 quickly for compromised clients. In contrast, the average trust score computed by FLTrust drops slowly. Second, the average genuine score computed by FedGame for benign clients first increases and then becomes stable. In contrast, the average genuine score computed by FLTrust for benign clients decreases as the number of iterations increases. As a result, FedGame outperforms FLTrust.

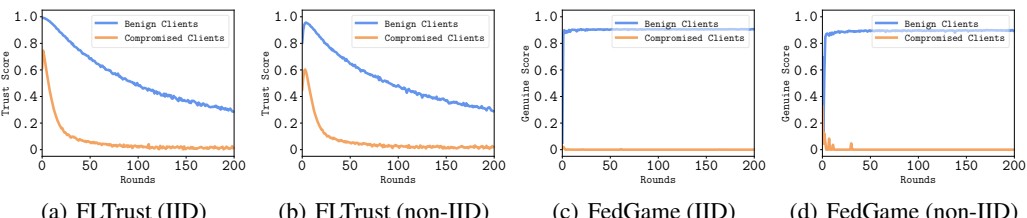

(a) FLTrust (IID)      (b) FLTrust (non-IID)      (c) FedGame (IID)      (d) FedGame (non-IID)

Figure 2: (a)(b): Server-computed average trust scores for benign and compromised clients of FLTrust on MNIST under Scaling attack. (c)(d): Average genuine scores computed by the server for benign and compromised clients of FedGame on MNIST under Scaling attack. The clean sets of the server are the same for FLTrust and FedGame.

## D.5 FedGame Performance in FL Consisting of 30 Clients

In Table 4, we report the performance of FedGame and baselines when the total number of clients is 30. The results also indicate that our FedGame outperforms all baselines in terms of ASR and achieves comparable TA with existing methods.

Table 4: Comparison of FedGame with existing defenses under Scaling attack. The total number of clients is 30 with 60% compromised. The best results when respectively comparing FedGame in each setting with existing defenses are bold.

| Datasets | Metrics | FedAvg (No attacks) | Defenses (Under attacks) | | | | | | | |
| | | | FedAvg | Krum | Median | Norm-Clipping | DP | FLTrust | FedGame In-domain | Out-of-domain |
|---|---|---|---|---|---|---|---|---|---|---|
| MNIST | TA (%) | 99.02 | 99.09 | 98.16 | 99.01 | 92.77 | 89.77 | 95.27 | 97.81 | 97.64 |
| | ASR (%) | 9.74 | 99.98 | 99.98 | 99.98 | 98.20 | 98.83 | 11.04 | **9.95** | **9.95** |
| CIFAR10 | TA (%) | 80.08 | 79.73 | 72.23 | 79.58 | 79.20 | 50.86 | 67.84 | 73.29 | 74.42 |
| | ASR (%) | 9.14 | 99.82 | 99.97 | 99.85 | 99.87 | 96.53 | 99.28 | **10.44** | **9.15** |

## D.6 Additional Ablation Studies

**Impact of Trigger Reverse Engineering.** Our framework is compatible with any trigger reverse engineer methods. To study the impact of different trigger reverse engineering methods, we use FeatureRE [47] to substitute Neural Cleanse in FedGame. We adopt the public implementation of FeatureRE. Under the default setting, FedGame achieves a 10.36% ASR, which indicates that our framework is consistently effective with different trigger reverse engineering methods.

Besides, FedGame would still be effective even if the reverse-engineered backdoor trigger and target label are of low quality, as long as they can distinguish benign and compromised clients. We validate this by experiment. In particular, we use 20 iterations (originally, it's 100) when reverse engineering the trigger/target label under the default setting to obtain a low-quality trigger/target label. We find that genuine scores for malicious clients are still low. The final ASR is 13.23%, which means our FedGame is still effective.

**Impact of the Total Number of Clients.** We study the impact of the total number of clients for our FedGame under the default setting. In particular, we consider the total number of clients to be 10, 30,

50, 70, and 100, where the fraction of malicious clients is 60%. We show the experimental results in Table 5. Our experimental results show that our FedGame is effective for different number of clients on different datasets.

Table 5: ASRs of FedGame under different total number of clients on MNIST and CIFAR10. The fraction of compromised clients is 60%.

| Dataset | Total Number of Clients | | | | |
|---|---|---|---|---|---|
| | 10 | 30 | 50 | 70 | 100 |
| MNIST | 9.72 | 9.95 | 10.03 | 10.01 | 9.89 |
| CIFAR10 | 8.92 | 10.44 | 10.62 | 9.79 | 10.82 |

**Impact of the Size of Server Clean Data.**  By default, we set the ratio between the number of clean examples of the server and the total number of examples of clients to be 0.1. We conduct experiments with different ratios: 0.01, 0.02, and 0.05 under the default setting. The corresponding ASRs are 9.71%, 12.38%, and 9.75%, indicating that FedGame is effective even when the server only has 1% clean data.

**Trigger Optimization.**  We consider an attacker optimizes trigger pattern such that a backdoored input is more likely to be predicted as the target class. We perform experiments under the default setting. The ASR is 12.43%, which indicates that our FedGame is consistently effective for trigger optimization.

