# OpenReview forum: "FedGame: A Game-Theoretic Defense against Backdoor Attacks in Federated Learning"
_NeurIPS.cc/2023/Conference — NeurIPS 2023 poster_

### Official Review · Reviewer_cSQx · 2023-07-04

**Soundness:** 2 fair
**Presentation:** 3 good
**Contribution:** 2 fair
**Rating:** 6
**Confidence:** 4

**Summary:**

This paper presents FedGame, a game-theory based dynamic defense mechanism against backdoor attacks in Federated Learning (FL). FedGame adapts to attackers using a minimax game model. The authors theoretically prove that a model trained with FedGame is close to one trained without attacks, implying robustness against malicious disruptions. Empirical evaluations on benchmark datasets reveal that FedGame compares with existing six existing defenses, reducing attack success rate on MNIST and CIFAR10.

**Strengths:**

1. Backdoor defense in federated learning is an important problem.
2. The paper provides theoretical analysis on the attack and defense as a two-player game.

**Weaknesses:**

1. The authors build an auxiliary global model to reverse engineer the trigger, as mentioned in Sec 4.2, to address the key challenge in solving Equation 5 is to construct an auxiliary global model. Then does the authors make any assumptions on the auxiliary global model $\theta^{t}_{a}$ comparing to global model? If so, is there any analysis on auxiliary global model in the minimax game framework? It is unclear that how the assumption is made and it would be beneficial to understand the implications if the assumed conditions do not hold true.
2. The determination of the 'genuine score', $p_{i}^{t}$, is another part that needs clarification. The paper does not provide theoretical estimations or robust definitions for such an important score metric for the defense, which raises questions about potential inaccuracies in the estimation and their impact on the defense framework.
3. The evaluation methodology could be further enhanced. If the genuine score is an empirical number, it would be useful to have it studied in the evaluation section. Moreover, comparisons with state-of-the-art baselines like FLIP [52] would provide a more comprehensive analysis of the proposed solution's performance.

**Questions:**

Please refer to the weaknesses.

**Limitations:**

Please refer to the weaknesses.

---

> ### Author Rebuttal · Authors · 2023-08-10
>
> We thank the reviewer for the constructive comments.
>
>
> **C1: Any assumption on the auxiliary global model and global model.**
>
> Thanks for the insightful question. Our auxiliary global model is obtained by updating the global model in the previous FL round with the average local model updates of the current FL round. Our assumption is that the auxiliary global model obtained in this way could be backdoored if a local model from a compromised client in the current FL round is backdoored, enabling us to reverse engineer the trigger to protect the current global model. In other words, if the auxiliary global model is not backdoored, our current global model would be very similar to the auxiliary global model and thus is also not backdoored. We will add related discussion and analysis in our revision.
>
> **C2: Potential inaccuracies in the estimation of the genuine score.**
>
> We note that our framework does not require a very accurate estimation of the genuine scores for clients, as long as the genuine scores are different for malicious clients (e.g., small genuine scores) and benign clients (e.g., large genuine scores). In Figure 2 of the Appendix, we show that the average genuine score for compromised clients is much smaller than that of benign clients, which validates the feasibility of the optimization problem. We will add related discussion in our revision.
>
> **C3: Genuine score should be studied in the evaluation section.**
>
> Thanks for the suggestion. In Figure 2 of the Appendix, we show the average genuine scores for compromised clients and benign clients. We will make it more clear in our revision.
>
> **C4: Comparison with baselines like FLIP.**
>
> Thanks for the suggestions. Following the suggestion, we compare our FedGame with FLIP under the default setting. We find that the FLIP can only reduce the ASR to 65.62\%, while our approach can reduce the ASR to 9.72\%. The reason is that their method can only tolerate a small fraction of compromised clients. In our default setting, the fraction of malicious clients is 60\%. We will add the related discussion and results in our revision.

---

> > ### Comment · Reviewer_cSQx · 2023-08-16
> >
> > Thanks for the response. I have raised my rating.

---

> > > ### Author Response · Authors · 2023-08-16
> > >
> > > We thank the reviewer for the insightful suggestions and positive feedback on our response!

---

### Official Review · Reviewer_8T34 · 2023-07-05

**Soundness:** 3 good
**Presentation:** 3 good
**Contribution:** 3 good
**Rating:** 6
**Confidence:** 3

**Summary:**

Existing defenses consider a static attack model where an attacker sticks to a fixed strategy and does not adapt its attack strategies, causing they are less effective under adaptive attacks. Hence, the authors propose FedGame to bridge this gap. The main idea of FedGame is to model single or multi-stage strategic interactions between the defender in FL and dynamic attackers as a min-max game. It mainly consists of three steps: 1) building an auxiliary global model. 2) exploiting it to reverse engineer a backdoor trigger and target class. 3) testing whether the local model will predict an input embedded with the reverse engineered trigger as the target class to compute a genuine score.

**Strengths:**

1. The idea of modeling strategic interactions between the defender and attackers as a min-max game is novel and less studied in previous works. The proposed FedGame seems to be competitive with related methods.
2. The paper is well-organized in general, well-written and mostly easy to follow.
3. This paper addresses a pertinent problem in Federated learning. The extensive experiments show that FedGame method works even if more than half of the clients are malicious.


**Weaknesses:**

1. The authors use the genuine score $p_i^t$ to quantify the extent to which a client is benign, however, the results of $p_i^t$ highly depends on the quality of reversed-trigger generated by NC or other tools, implying the limitations of the FedGame are the same as NC, e.g., maybe invalid in large-size trigger reverse (such as Blend, Reflection, …) and all-to-all attacks, etc.
2. FedGame seems to have high time complexity. In the server side, the defender needs to run NC process to generate reverse-trigger (When there are many categories, the time consumption of NC is enormous). In the client side, the attacker needs to find an optimal $r_i^t$ using grid search (Every search for $r_i^t$ requires retraining the local model).



**Questions:**

1. The paper points out that existing defenses become less effective against dynamic attackers who strategically adapt their attack strategies. However, the definition of dynamic attacks is unclear. Further, the dynamic attack simulated by this paper seems to be only the change of the poisoning rate $r_i^t$. Does this mean that the proposed method in this paper has certain limitations? Because there are many possible factors that may affect the attack performance, such as the selection of poisoned neurons [1], the design of the loss function [2], etc.
2. In practical, the defender cannot manipulate the attacker's malicious behavior. Hence, it seems the defense method proposed in this paper just need to achieve the goal of $\min (\sum_{i \in S_a} p_i + \sum_{j \in S-S_a} p_j)$. If so, the FedGame is more likely a combination of NC method with limited impact from the min-max game. If not, does it mean that FedGame needs benign clients to act as attackers to achieve better defense effects? Will FedGame be invalid if the target of the real attackers and the fake attackers are different?
3. In section 4.3, when searching for an optimal $r_i^t$, the goal of attackers is $\max (p_i^t+\lambda r_i^t)$. However, increasing $p_i^t$ and increasing $r_i^t$ are contradictory because large $p_i^t$ means small ASR, while large $r_i^t$ means high ASR. It seems that the attack constructed in this paper is not competitive enough, which raises some doubts about the necessity of the attack stage.


[1]. PerDoor: Persistent Non-Uniform Backdoors in Federated Learning using Adversarial Perturbations.
[2]. Chameleon: Adapting to Peer Images for Planting Durable Backdoors in Federated Learning.


**Limitations:**

see the questions.

---

> ### Author Rebuttal · Authors · 2023-08-10
>
> We thank the reviewer for the constructive comments.
>
> **C1: Large-size trigger.**
>
> We note that our framework is very general and is compatible with any trigger reverse engineering methods such as NC and FeatureRE. For instance, FeatureRE [42] (published in NeurIPS’22) shows that it is effective for large-size triggers (even for triggers that occupy the entire image), which addresses the limitations of NC. We will make it more clear and add related discussion in our revision.
>
> **C2: Time complexity.**
>
> We agree with the reviewer that our defense incurs extra computation costs for the server. However, we note that the server (e.g., servers in the FL applications deployed by Google and Apple) is usually very powerful in practice. Moreover, the reverse engineered triggers could be used for all clients at once. As we propose a defense framework, we consider a strong attacker, where a compromised client could use grid search to find an optimal $r_i^t$ to maximize the attack's effectiveness. The attack would be less effective if the compromised client cannot optimize $r_i^t$. We will add the related discussion.
>
> **C3: The definition of dynamic attacks is unclear.**
>
> Sorry for the confusion. In our dynamic attack, we consider an attacker could optimize both the poisoning rate and trigger pattern. In Appendix D.6, our experimental results show our defense is still effective when an attacker can optimize the entire trigger pattern. We will clarify it in our revision. Moreover, we will discuss other factors such as the selection of poisoned neurons [1] and the design of the loss function [2] as suggested. In addition, following the suggestions, we have added an experiment which uses the loss function in [2]. We observe that our method can reduce the ASR from 98.63\% to 11.32\%, indicating the effectiveness of our method.
>
> **C4: Defense goal.**
>
> Yes, the goal of the defender is to minimize (or maximize) the genuine scores for compromised (or benign) clients, i.e., $\min (\sum_{i \in S_a} p_i - \sum_{j \in S -S_a} p_j)$. Since the benign clients do not have attack behavior, genuine scores obtained by our framework for benign clients would be high, enabling us to distinguish benign and compromised clients. Note that our major contribution is to propose a general game-theory based framework, which is compatible with any trigger reverse engineering method such as NC and FeatureRE. Our results show our framework is also effective with FeatureRE. We will add the related discussion.
>
> **C5: Optimization of $r_i^t$ and $p_i^t$.**
>
> We note that the local model of a compromised client is more likely to predict a trigger-embedded input as the target class when the poisoning rate $r_i^t$ is larger. As a result,  the genuine score computed by the server for the local model of the compromised client would be smaller, making the attack less effective. In other words, a larger $r_i^t$ makes the attack more effective without defense but less stealthy (e.g., the genuine score is smaller under our defense framework), which makes the attack less effective under defense, and this forms the defense principle. Thus, we aim to find an optimal $r_i^t$ to achieve a tradeoff between the poisoning rate and the genuine score to maximize the attack effectiveness under our defense. We will add related discussion to make it more clear in our revision.

---

### Official Review · Reviewer_Mtx6 · 2023-07-07

**Soundness:** 3 good
**Presentation:** 3 good
**Contribution:** 3 good
**Rating:** 6
**Confidence:** 2

**Summary:**

Federated learning (FL) enables a distributed training paradigm, where multiple clients can jointly train a global model without needing to share their local data. However, recent studies have shown that federated learning provides an additional surface for backdoor attacks. For instance, an attacker can compromise a subset of clients and thus corrupt the global model to mispredict an input with a backdoor trigger as the adversarial target. Existing defenses for federated learning against backdoor attacks usually detect and exclude the corrupted information from the compromised clients based on a static attacker model. Such defenses, however, are not adequate against dynamic attackers who strategically adapt their attack strategies. To bridge this gap in defense, we model single or multi-stage strategic interactions between the defender in FL and dynamic attackers as a minimax game. Based on the analysis of our model, we design an interactive defense mechanism FedGame. We also prove that under mild assumptions, the global FL model trained with FedGame under backdoor attacks is close to that trained without attacks. Empirically, we perform extensive evaluations on benchmark datasets and compare FedGame with multiple state-of-the-art baselines. Our experimental results show that FedGame can effectively defend against strategic attackers and achieves significantly higher robustness than baselines. For instance, FedGame reduces attack success rate by 82% on CIFAR10 compared with six state-of-the-art defense baselines under Scaling attack.

**Strengths:**

The article is well written, has a smooth grammar, and a clear purpose. It better summarizes and clarifies the concepts and definitions of backdoor attack and Federated learning. It is novel and innovative to express FedGame as a minimax game between defenders and attackers.The mathematical derivation is relatively detailed.

**Weaknesses:**

There may be biases in the selection and collection of data samples, which affect the accuracy and reliability of the results. In terms of research methods, there is a lack of sufficient description of the control group and experimental group, making it difficult to determine causal relationships. The literature review section lacks comprehensive and in-depth academic research, resulting in a lack of clarity in the research background of the article. The discussion on the results and discussion section is not clear and powerful enough, and requires more detailed explanation and support. The improper use of statistical analysis methods or hypothesis testing has affected the credibility of the results. There are some loose and incoherent issues in the structure and organization of the paper, which require better logic and framework. The summary in the conclusion section may be too brief to fully summarize the significance and contributions of the research.The writing of the paper could be improved for better description and clarification.

**Questions:**

1.Have you cited the latest and relevant research results？
2.Insufficient experimental content.
3.The experimental datasets are all small datasets, and model testing needs to be carried out on the Big data set.
4.Icons are not clear enough


**Limitations:**

The author's paper has good innovation and rigorous writing logic, but the experiment is not sufficient and needs to be further expanded to obtain more universal conclusions.

---

> ### Author Rebuttal · Authors · 2023-08-10
>
> We thank the reviewer for the constructive comments.
>
> **C1: Have you cited the latest and relevant research results?**
>
> Thanks for the question. To the best of our knowledge, we have cited those research results. We will do a comprehensive literature survey to add and discuss more recent research results (e.g., those published after the NeurIPS’23 deadline). Thanks for the kind reminder!
>
> **C2: Insufficient experimental content.**
>
> Following the suggestions, we will add more discussion and explanation in our experimental results sections. We will also polish the structure and organization of the paper. Moreover, we will expand the conclusion section and polish the writing of the paper to make it clear in the description.
>
> **C3: The experimental datasets are all small datasets.**
>
> In Appendix D.6, we conduct experiments on the Tiny-ImageNet dataset. Our experimental results show our defense is still effective on the large dataset. We will make it more clear in our revision.
>
> **C4: Icons are not clear enough.**
>
> Thanks for the suggestions. We will polish the figures in the paper to make them clear.

---

> > ### Comment · Reviewer_Mtx6 · 2023-08-15
> >
> > Dear authors: I extend my gratitude for the comprehensive responses and the inclusion of additional experiments. Overall, I find myself without any further inquiries concerning this manuscript.

---

> > > ### Author Response · Authors · 2023-08-15
> > >
> > > We really appreciate the reviewer for the time and effort in reviewing the paper and reading our rebuttal.

---

### Official Review · Reviewer_89HC · 2023-07-21

**Soundness:** 2 fair
**Presentation:** 2 fair
**Contribution:** 3 good
**Rating:** 5
**Confidence:** 4

**Summary:**

The authors introduce a game-theoretical defense mechanism named "FedGame" designed to safeguard against backdoor attacks in the Federated Learning (FL) environment. They demonstrate through theoretical analysis that training the global FL model using FedGame under backdoor attacks yields results closely resembling those obtained without any attacks. The researchers conduct extensive evaluations using benchmark datasets and compare FedGame's performance with various state-of-the-art baselines. The experimental findings indicate that FedGame successfully thwarts strategic attackers and significantly outperforms the baseline approaches, displaying remarkable robustness against adversarial threats.

**Strengths:**

1. It is interesting to incorporate game theory to federated learning.
2. The authors show that using FedGame to train the global FL model under backdoor attacks yields similar results to those without attacks.
3. The authors extensively evaluate FedGame on benchmark datasets, showcasing its effectiveness in defending against strategic attackers and outperforming baselines with significantly higher robustness.

**Weaknesses:**

From game perspective,
1. Ideally, we expected to see a game model incorporating FL's online nature (e.g., Markov Game). It is unclear to me why it is a multi-stage strategic game (sequential decision problem) since each FL round's optimization problem is solved independently.
2. The term minmax game is ambiguous. It should be making clear which solution concept (Stackelberg or Nash equilibrium, Zero-sum or General-sum utility) is considered. Otherwise, it is hard to claim the existence of the solution and convergence of the algorithm.
3. The game's information design is unclear (game with complete or incomplete information). It should be modeled clearly (probably mathematically?) which player have what information that is known/unknown by some other player. Informational concerns play a central role in players’ decision making in such strategic environments.

From security perspective,
1. Why the reverse engineering methods can be directly applied to FL setting considering they normally require large amount of clean data without bias (FL clients may hold heterogeneous data)? Will the aggregation influence the reversed pattern since the aggregated model is updated by a combination of clean and poisoned data?
2. The action spaces for both attacker and defender are not general enough. From the attack side, poison ratio is not the only parameter the attacker can manipulate (e.g., scaling factor, batch size, number of local training epoch, etc.). In general, attacker can control the model updates send to the server. From the defense side, the authors only consider the client-wise aggregation-based training stage defense. There are other defense (e.g., coordinate-wise, post training stage defenses) could be considered to extent the action space.
3. In FL, sever is not only unaware of the attacker, but also lack of environment information, e.g., number of total clients, non-iid level, subsampling, etc.

**Questions:**

1. Instead of utilizing reverse engineering, why not considering other approaches closer to game concept like modeling a Bayesian game, and update defender's prior belief about attacker during FL process?
2. Does the "dynamic backdoor attack" mean the attacker can change backdoor trigger and targeted label during FL process? If that is the case, the attacker should consider a long-term goal instead of a one-shot myopic objective. Otherwise, it could mitigate its own attack effects even there is no defense exists.
3. It is unclear to me in both section 4.3 and Appendix how the attacker uses stochastic gradient descent to optimize the trigger $\delta$. In the conducted DBA experiments, the trigger(s) take the form of fixed-sized square(s) located at specific position(s) and may involve certain numbers of squares. These characteristics represent potential parameters that can be incorporated into the optimization problem. However, it is worth noting that for achieving the most versatile trigger, it is worthwhile to consider every pixel in the image. In recent works [1] [2], researchers have explored generated backdoor triggers that encompass the entire range of the image.
4. I wonder what if the reverse engineering cannot get the correct trigger and/or targeted label? There should be either theoretical analysis or ablation experiments to further exam the influence.


[1] Salem, Ahmed, et al. "Dynamic backdoor attacks against machine learning models." 2022 IEEE 7th European Symposium on Security and Privacy (EuroS&P). IEEE, 2022.
[2] Doan, Khoa D., Yingjie Lao, and Ping Li. "Marksman backdoor: Backdoor attacks with arbitrary target class." Advances in Neural Information Processing Systems 35 (2022): 38260-38273.

**Limitations:**

1. The post-training stage defenses play a vital role in countering backdoor attacks. Even within the context of FL, certain techniques such as Neuron Clipping [3] and Pruning [4] have demonstrated their effectiveness in detecting and mitigating the impact of backdoor attacks. Consequently, I am curious to know how the proposed FedGame performs compare to these post-training stage defenses. Noted that, some of them do not require to know backdoor trigger or target label.
2. I wonder the efficiency of the FedGame algorithm. It seems for each FL epoch, it needs process reverse engineering method (costly according to original papers) and calculate a equilibrium (also costly in most of settings like genera sum Nash aquarium). Is there any assumption or simplification I missed here?

[3] Wang, Hang, et al. "Universal post-training backdoor detection." arXiv preprint arXiv:2205.06900 (2022).
[4] Wu, Chen, et al. "Mitigating backdoor attacks in federated learning." arXiv preprint arXiv:2011.01767 (2020).

---

> ### Author Rebuttal · Authors · 2023-08-10
>
> We thank the reviewer for the valuable suggestions.
>
> **C1: Why the game is multi-stage.**
>
> We are sorry for the confusion. In the abstract, we meant to say a multi-stage interaction between the attacker and defender over different FL rounds instead of a multi-stage game.
>
> **C2: Minmax term not clear.**
>
> We are sorry for the confusion. Our game is a zero-sum Stackelberg game. We will add the discussion of the existence of the Stackelberg equilibrium following existing literature.
>
> **C3: Game’s information.**
>
> The defender knows local models but does not have 1) information on compromised clients, 2) poisoning rate $r_i^t$, trigger $\delta$, and target class $y^{tc}$ of the attacker. The attacker knows the global model $\Theta^t$ in each FL round and the trigger reverse engineer method (to consider a strong attacker). We will add more details.
>
> **C4: How does reverse engineering in FL work?**
>
> We would like to kindly mention that we don't heavily rely on the resources of clean data. FedGame does not require reverse-engineering the exact trigger as long as the reverse-engineered trigger can be used to distinguish benign and compromised clients. Thus, compared with [38, 42], we have a lower requirement for the reconstructed trigger. In our experiments, we only use a moderate number of clean samples (10% the size of training data in the default setting) to reverse engineer. The aggregation step will not affect the effectiveness of reverse engineering, since these FL attacks are designed to poison the global model with only a few local models poisoned.
>
> **C5: Action space.**
>
> We really appreciate the suggestion. We hope to provide a general framework. We conduct thorough experiments considering the poisoning ratio as the action space of players since it is critical in controlling the effectiveness and cost of attackers. Moreover, we also consider optimizing other variables, such as the trigger pattern. We agree that it is an interesting future work to extend the action space. We will discuss.
>
> **C6: Environment information.**
>
> We agree. Note that we don’t assume the server is aware of the environment information to ensure the generalizability of FedGame. For instance, without an attacker, FedGame would automatically assign similar genuine scores for clients, reducing FedGame to FedAvg, which often gives the best utility. With an attacker, FedGame still does not need to know the number of total clients, non-iid level, or subsampling, e.g., our results show FedGame is effective with 1) different total number of clients, and 2) a subset of clients was selected in each FL round. We will make this clear.
>
> **C7: Other game formulations.**
>
> Here we aim to propose the first game-theory-based framework. We believe it is an interesting future direction to consider other game formulations, e.g., Bayesian games. We will add a discussion.
>
> **C8: Dynamic backdoor attacks.**
>
> Yes. We consider and allow a dynamic attacker who could change the backdoor trigger and targeted label during the FL process. Currently, we only consider the possibly worse-case attacker in each round by optimizing their attack objective. We agree that an attacker could consider a long-term goal, leading to more sophisticated attacks. We leave the attack development (developing such an attack is still an open challenge) and the examination of FedGame under the attack as future work. We will add related discussion and future work.
>
> **C9: Trigger optimization.**
>
> We are sincerely sorry for the confusion. We optimize the trigger according to different attacks. For instance, we follow the setting of DBA, and optimize the position for each local trigger and preserve its pattern. For scaling attack, we optimize every pixel of the trigger. Concretely, we randomly sample a batch of samples from $\mathcal{D}_i$, then compute the gradient of the loss function in Line 219 with respect to $\delta^*$, and finally update $\delta^*$ based on the gradient. Note that we clip the gradient such that each entry of $\delta^*$ is a valid pixel value. We will add detailed illustrations.
>
> As suggested, we added an experiment for triggers in [1]. FedGame reduces ASR from 99.43% to 10.83% under our default setting, meaning FedGame is effective for diverse triggers. We will add results and discussion.
>
> [1] "Dynamic backdoor attacks against machine learning models."
>
> **C10: Reverse-engineered trigger/targeted label could be incorrect.**
>
> FedGame would still be effective if reconstructed triggers were incorrect, as long as they can distinguish benign and compromised clients. We added an experiment to validate this. We use 20 iterations (originally, it’s 100) when reverse engineering the trigger under the default setting to obtain low-quality and incorrect triggers (the trigger is not visually similar to the original one in this case). We find that genuine scores for malicious clients are still low. The final ASR is 13.23% which is slightly higher than the original ASR. We will add a discussion.
>
> **C11: Post-training defenses.**
>
> FedGame could be combined with those defenses to form a defense-in-depth. To maintain utility, in general, [3] needs some clean samples that have the same distribution as the training dataset, while FedGame is effective even if samples of server have different distributions from FL task. [4] prunes filters based on pruning sequences collected from clients. Thus, [4] could be less effective under a large fraction of compromised clients, while FedGame is effective under 80% of compromised clients. We will add the discussion on [3, 4] and comparison.
>
> **C12: Efficiency.**
>
> We agree that FedGame incurs extra computation costs for the server. We would like to kindly note that in practice, the server (e.g., deployed by Google) is usually powerful. Given the triggers, it is very efficient to compute genuine scores for clients, as shown in Appendix D.6. Moreover, we could compute them in parallel. We will add these discussions in our revision.

---

> > ### Comment · Reviewer_89HC · 2023-08-17
> >
> > I appreciate the authors' response and informative clarification. After reading the rebuttal and other reviewers' comments, most of my concerns have been addressed. I will change my rating. Thank you.

---

> > > ### Author Response · Authors · 2023-08-17
> > >
> > > We really appreciate the reviewer for the positive feedback on our response and constructive suggestions!

---

### Author Rebuttal · Authors · 2023-08-10

We really appreciate the reviewers for the constructive feedback and suggestions. We are glad that the reviewers find our work novel, effective, and well-organized. In summary, we have added the following additional experiments following the reviewers’ suggestions:

1. We add an experiment to validate the effectiveness of FedGame for triggers that encompass the entire range of the image.
2. We add an ablation experiment to show our FedGame is still effective when reverse engineered trigger/targeted label could be incorrect.
3. We add an experiment to validate the effectiveness of FedGame when an attacker could design a different loss function.
4. We add a comparison with FLIP (a defense baseline).



In addition, we have added the following clarifications/explanations in our revision:

1. We clarify the minmax game of our defense.
2. We clarify the information design in our game.
3. We add an explanation on the trigger reverse engineering in FL.
4. We clarify the environment setup.
5. We clarify the dynamic backdoor attack strategies.
6. We add the details of the trigger optimization.
7. We clarify and discuss the post-training defenses.
8. We discuss the efficiency of our framework.
9. We clarify the relevant research and experiment content.
10. We add an explanation of our framework for large triggers.
12. We add an explanation on the optimization of $r_i^t$ and $p_i^t$.
13. We clarify the auxiliary global model and global model.
14. We add an explanation on genuine score estimation.
15. We clarify the evaluation steps of the genuine score.

---

### Decision · Program_Chairs · 2023-09-21

**Decision:**

Accept (poster)

**Comment:**

All reviewers agree that expressing FedGame as a minimax game between defenders and attackers is a novel idea. The authors did an excellent job addressing most reviewers' concerns during the rebuttal. After the discussion phase, all reviewers recommended acceptance. The Area Chair recommends accepting the paper. In the final version, the authors should address the reviewers' critical comments. These include the effect of the quality of the reversed trigger on the method and a more detailed discussion about the extra computational costs at the server.